# Development of photosynthetic carbon fixation model using multi-excitation wavelength fast repetition rate fluorometry in Lake Biwa

Takehiro Kazama[1,2]*, Kazuhide Hayakawa[3], Victor S. Kuwahara[4], Koichi Shimotori[1,2], Akio Imai[1], Kazuhiro Komatsu[2]

**1** Lake Biwa Branch Office, National Institute for Environmental Studies, Otsu, Shiga, Japan, **2** Center for Regional Environmental Research, National Institute for Environmental Studies, Tsukuba, Ibaraki, Japan, **3** Lake Biwa Environmental Research Institute, Otsu, Shiga, Japan, **4** Graduate School of Science & Engineering, Soka University, Hachioji, Tokyo, Japan

* kazama.takehiro@nies.go.jp

## Abstract

Direct measurements of gross primary productivity (GPP) in the water column are essential, but can be spatially and temporally restrictive. Fast repetition rate fluorometry (FRRf) is a bio-optical technique based on chlorophyll $a$ (Chl-$a$) fluorescence that can estimate the electron transport rate ($ETR_{PSII}$) at photosystem II (PSII) of phytoplankton in real time. However, the derivation of phytoplankton GPP in carbon units from $ETR_{PSII}$ remains challenging because the electron requirement for carbon fixation ($\Phi_{e,C}$), which is mechanistically 4 mol $e^-$ mol $C^{-1}$ or above, can vary depending on multiple factors. In addition, FRRf studies are limited in freshwater lakes where phosphorus limitation and cyanobacterial blooms are common. The goal of the present study is to construct a robust $\Phi_{e,C}$ model for freshwater ecosystems using simultaneous measurements of $ETR_{PSII}$ by FRRf with multi-excitation wavelengths coupled with a traditional carbon fixation rate by the $^{13}C$ method. The study was conducted in oligotrophic and mesotrophic parts of Lake Biwa from July 2018 to May 2019. The combination of excitation light at 444, 512 and 633 nm correctly estimated $ETR_{PSII}$ of cyanobacteria. The apparent range of $\Phi_{e,C}$ in the phytoplankton community was 1.1–31.0 mol $e^-$ mol $C^{-1}$ during the study period. A generalised linear model showed that the best fit including 12 physicochemical and biological factors explained 67% of the variance in $\Phi_{e,C}$. Among all factors, water temperature was the most significant, while photosynthetically active radiation intensity was not. This study quantifies the $in\ situ$ FRRf method in a freshwater ecosystem, discusses core issues in the methodology to calculate $\Phi_{e,C}$, and assesses the applicability of the method for lake GPP prediction.

**Data Availability Statement:** All relevant data are within the paper and its Supporting Information files.

**Funding:** TK, KH, KS and AI were supported by the Collaborative Research Fund from Shiga Prefecture entitled "Study on water quality and lake-bottom environment for protection of the soundness of water environment" under the Japanese Grant for Regional Revitalization, and the Environment Research and Technology Development Fund (No. 5-1607) of the Ministry of the Environment, Japan. https://www.kantei.go.jp/jp/singi/tiiki/tiikisaisei/souseikoufukin.html.

**Competing interests:** The authors have declared that no competing interests exist.

# Introduction

Phytoplankton are the most important primary producers in the aquatic food web [1]. Changes in phytoplankton primary productivity can affect the food chain length [2, 3], material cycles [4, 5] and biomass of higher trophic organisms [6–8]. Phytoplankton community productivity is affected by various environmental factors and must rapidly respond to them [9–11] due to high growth rates and short generation times [12]. To evaluate the effect of variability in environmental factors on aquatic communities and ecosystems, continuous observation of phytoplankton primary productivity is necessary [4, 7, 13]

Traditional chemical methods of measuring primary production, such as the $^{14}C$ method [14], the $^{13}C$ method [15, 16], the light-dark bottle method [17] and the $^{18}O$ method [18], require handling a radioisotope ($^{14}C$) and/or an incubation time of several hours. Thus, primary production studies using these techniques can be limiting when attempting to assess temporal and spatial variability. Fast repetition rate fluorometry (FRRf; Table 1), a chlorophyll *a* fluorescence-based method, has been developed as an advanced bio-optical technique for real-time measurement of phytoplankton primary productivity, mainly in marine ecosystems [19–27]. The FRRf method enables the induction and measurement of a range of chlorophyll *a* fluorescence yields and parameters specific to photosystem II (PSII) [19, 20, 28], and, in turn, enables estimation of the *in vivo* electron transport rate in PSII ($ETR_{PSII}$) and gross primary productivity (GPP) by theoretical models of photosynthesis [19, 28, 29].

Previous studies demonstrated that GPP estimated from FRRf measurements correlated well with results from conventional methods, including the $^{13}C$ method [25, 26, 30, 31] and the light-dark bottle method [24]. However, FRRf measurements tended to overestimate GPP compared with the $^{14}C$ and the $^{13}C$ methods, while underestimating them when compared with the light-dark bottle and the $^{18}O$ methods [32]. These discrepancies in GPP measurements are dependent on the targeted products, namely, oxygen or particulate organic carbon, in the photosynthesis cycle [33]. To account for the measurement discrepancies, recent studies have examined the electron requirement for carbon fixation ($\Phi_{e,C}$, also called *Kc*) by comparing the FRRf-derived $ETR_{PSII}$ per unit volume ($JV_f$) to the GPP rate by traditional methods [25–27, 34]. The $\Phi_{e,C}$ is affected by multiple spatiotemporal variations in physical and chemical factors [35–40], and by phytoplankton community composition [26, 41–44]. For example, $\Phi_{e,C}$ is higher in the open ocean than in coastal areas due to differences in the light environment conditions; light availability is higher in open oceans [35]. Light availability can be affected by the concentrations of colored dissolved organic matter (CDOM), non-algal particles, and phytoplankton. Moreover, the vertical distribution of phytoplankton is markedly different in open-ocean and coastal areas, where the former has peak biomass located in a much deeper area with less CDOM and non-algal particles in the water column, allowing more light availability [35]. More specifically, excess light energy enhances photo-oxidative damage and alternative electron transport such as the Mehler reaction, flavodi-iron protein-mediated electron flows and the plastiquinol terminal oxidase (PTOX) pathway, which can increase $\Phi_{e,C}$ [33, 45]. In addition to ambient light conditions, nutrients can also play an important role in determining $\Phi_{e,C}$ of the phytoplankton community [27, 39]. For example, Schuback et al. [27] described the negative relationships between $\Phi_{e,C}$ and nitrate concentrations in the Arctic Ocean, suggesting the variable effects of nutrient availability. The multitude of interacting factors affecting the value of $\Phi_{e,C}$ for converting $JV_f$ to GPP make it difficult to establish a general model applicable to different ecosystems [35]. Therefore, to construct a robust $JV_f$-GPP model, it is necessary to accumulate FRRf and its corresponding GPP data in various aquatic environments.

**Table 1. Terms used within this manuscript.**

| Term | Definition | Units |
|---|---|---|
| $a$ ($_{CDOM, NAP,}$ $^{*}_{phy}$ or $_w$) | Absorption spectrum (of CDOM, NAP, phytoplankton or pure water) | m$^{-1}$ or m$^{-2}$ mg Chl-$a^{-1}$ |
| AEF | Alternative electron flow | |
| $E$ ($_{FRRf,\ 0,}$ or $_{in\ situ}$) | Incident photosynthetically active radiation between 400 and 700 nm (of excitation flashlets, or at 0, or *in situ*) | μmol photons m$^{-2}$ s$^{-1}$ |
| ETR$_{(PSII)}$ | Electron transport rate (in PSII) | |
| FRRf | Fast Repetition Rate Fluorometry | |
| GLM | Generalized linear model | |
| GPP | Gross primary productivity per area | g C m$^{-2}$ d$^{-1}$ |
| $\Phi_{e,C}$ | Electron requirement for carbon fixation, also called $Kc$ | mol e$^-$ mol C$^{-1}$ |
| $K_d$ | Extinction coefficient of light | m$^{-1}$ |
| PSII | Photosystem II | |
| RCII | PSII reaction centers | nmol m$^{-3}$ |
| SCF | Spectral correction factor | |
| **Fluorometry** | | |
| $C$ | Fraction of RCII in closed state | |
| $F'$ | Fluorescence at zeroth flashlet of a single turnover measurement when $C>0$ | |
| $F_O$ (') | Minimum PSII Fluorescence yield (under acclimation to background light) | |
| $F_v$ (') | $F_m(') - F_O(')$ | |
| $F_m$ (') | Maximum PSII Fluorescence yield (under background light) | |
| $F_v/F_m$ | $(F_m - F_O)/F_m$ | |
| $GPP_f$ | GPP estimated by FRRf | g C m$^{-2}$ d$^{-1}$ |
| $J_f$ | RCII-specific electron transport rate in PSII based on FRRf | μmol e$^-$ nmol RCII$^{-1}$ s$^{-1}$ |
| $JV_f$ | Electron transport rate per water volume in PSII based on FRRf | μmol e$^-$ m$^{-3}$ s$^{-1}$ |
| $NPQ_{NSV}$ | Non-photochemical quenching based on normalized Stern-Volmer quenching coefficient | |
| $qP$ | Separate package model of connectivity between RCIIs | |
| $R\sigma_{PSII}$ (') | Probability of an RCII being closed during the first flashlet of a single turnover saturation phase (under background light) | |
| $\sigma_{PSII}$ (') | Functional absorption cross section of PSII for excitation flashlets (under background light) | nm$^2$ |
| **$^{13}$C method** | | |
| $GPP_{13C}$ | GPP estimated by $^{13}$C | g C m$^{-2}$ d$^{-1}$ |
| $GP_C$ | Gross primary productivity per water volume based on $^{13}$C | mg C m$^{-3}$ h$^{-1}$ |
| $PB_C$ | RCII-specific primary productivity based on $^{13}$C | mg C nmol RCII$^{-1}$ h$^{-1}$ |
| **Light-dark bottle method** | | |
| $JV_O$ | Electron transport rate per water volume based on light-dark bottle | μmol e$^-$ m$^{-3}$ s$^{-1}$ |
| $NPo$ | Net oxygen evolution rate based on light-dark bottle | mg O$_2$ m$^{-3}$ h$^{-1}$ |
| $R_d$ | Dark respiration rate based on light-dark bottle | mg O$_2$ m$^{-3}$ h$^{-1}$ |

In terms of physicochemical and biological conditions, freshwater ecosystems differ considerably from marine ecosystems. For example, cyanobacteria (blue-green algae) can frequently form dense surface blooms in meso–eutrophic lakes [46, 47]. Previous studies have suggested that cyanobacterial blooms can significantly affect ETR$_{PSII}$ measurements due to spectral mismatch between FRRf excitation wavelengths and the absorption spectrum of cyanobacteria [21, 26, 36, 41, 48]. For example, Raateoja et al. [41] found that filamentous cyanobacteria *Nodularia*

*spumigena* and *Aphanizomenon* sp. had absorption peaks around 630 nm, and the photosynthetic activity of these species could not be measured by FRRf with an excitation light around 475 nm (targeting Chl-*a*). Cyanobacteria have multiple absorption peaks around 500–570 nm and 630 nm based on antenna pigments [49]. Moreover, the blue light ($<$ 500 nm) absorbed by cyanobacterial antenna pigments does not make a major contribution to the reduction of the $Q_A$ and PQ pool, and $O_2$ evolution [50]. Thus, it is critical that the FRRf excitation wavelengths correspond to the absorption spectrum of cyanobacteria (or the dominant group) for accurate estimation of primary productivity and model development in freshwater ecosystems [21, 41, 51].

Phytoplankton primary productivity is also more likely to be phosphorus-limited in freshwater ecosystems [52, 53], while more likely to be nitrogen-limited in marine environments [54, 55]. Whereas nitrogen limitation depresses cellular Chl-*a* concentration, phosphorus limitation inhibits RNA and ATP synthesis, which can affect protein synthesis and photochemical energy conversion in algae. [56, 57]. In fact, $\Phi_{e,C}$ of marine phytoplankton increases under nitrogen limitation [27, 39]. Regrettably, the influence of phosphorus stoichiometry on $\Phi_{e,C}$ in freshwater phytoplankton remains unknown, and FRRf studies in freshwater environments in general are still limited [58–60]. Owing to the differences between marine and freshwater ecosystems, it is essential to employ suitable excitation wavelength combinations and measure phosphorus concentration in the water column to correctly estimate both the $ETR_{PSII}$ and GPP of freshwater phytoplankton communities.

The goal of the present study is to construct a robust $\Phi_{e,C}$ model applicable to freshwater ecosystems using simultaneous measurements of $ETR_{PSII}$ by FRRf coupled with the traditional carbon fixation rate by the $^{13}$C method in Lake Biwa. In this context, we first evaluated the performance of FRRf with multi-excitation wavelengths (444, 512, and 633 nm; S1A Appendix) during cyanobacterial blooms. Then, we evaluated the relative importance of variable environmental and biological factors, including phosphorus concentration in the water column, by determining $\Phi_{e,C}$ by statistical models with multiple variables. This study shows the utility of *in situ* FRRf measurements using an excitation wavelength of 633 nm during cyanobacterial blooms, and the extent to which physicochemical factors and phytoplankton community composition influence $\Phi_{e,C}$ estimation.

## Materials and methods

### Ethics statement

No permits were required for the described study, because the location was not privately-owned or protected, and the field studies did not involve endangered or protected species.

### Study site

The study was conducted at Lake Biwa (670 m$^2$ surface area with a mean depth of 43 m) on Honshu Island, Japan (Fig 1). The North Basin is a deep, oligotrophic area, while the South Basin is a shallow, mesotrophic area [61]. Phytoplankton communities differ markedly between the two basins, especially in terms of cyanobacterial abundance [62]. Sampling was carried out at the long-term survey stations, 12B (62 m depth) and 17B (89 m depth) in the North Basin, and at 6B (4 m depth) and 9B (5.6 m depth) in the South Basin, from July 2018 to May 2019 (Table 2). No permits were required to sample the lake in this study.

### Sampling procedure

Vertical profiles of water temperature, pH, dissolved oxygen concentration, and specific conductivity were measured using a water quality sonde (EXO2; Xylem, Inc., Yellow Springs, OH,

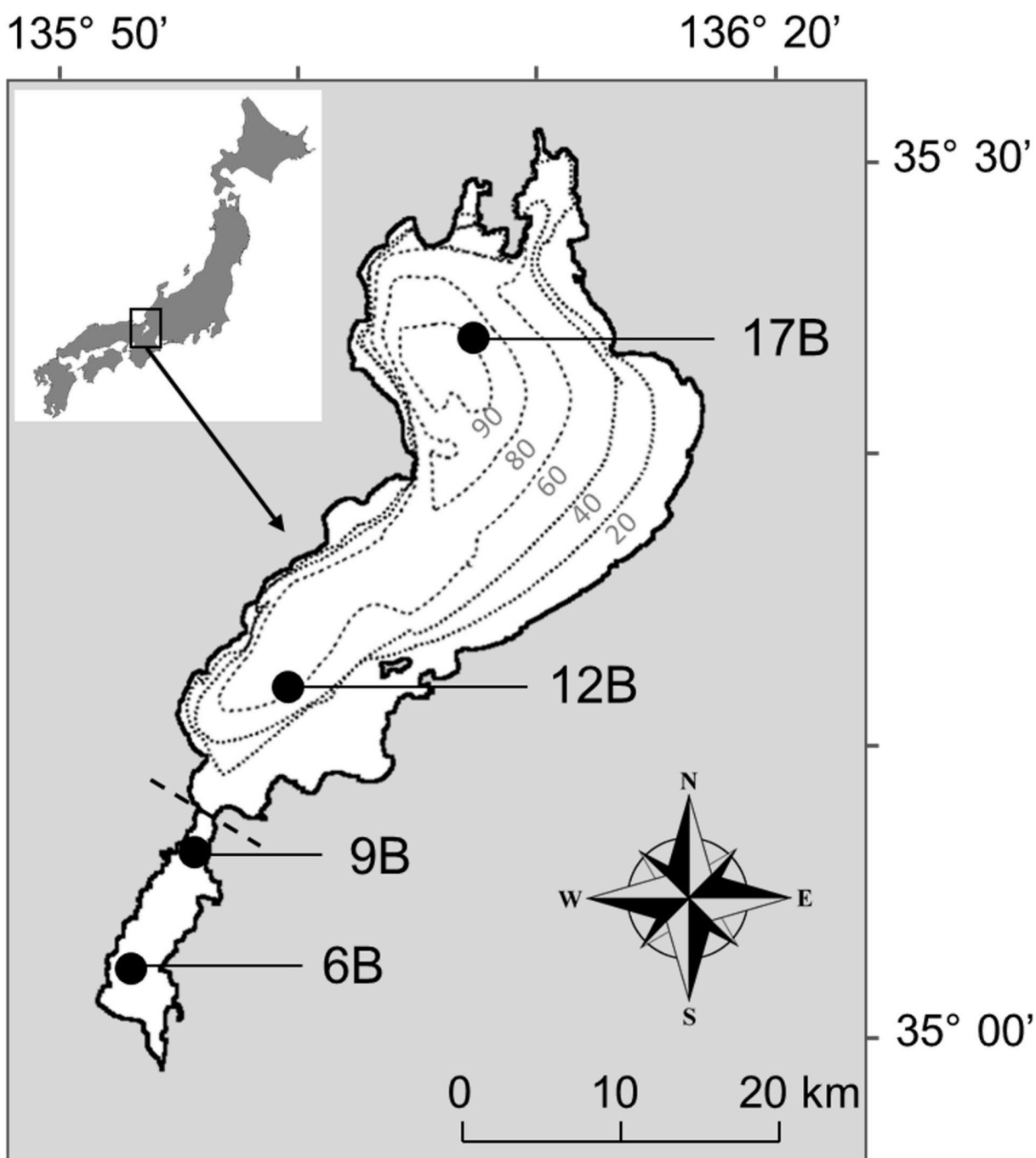

**Fig 1. Map of study sites in Lake Biwa, Japan.** Stations 6B and 9B represent the South Basin, while Stations 12B and 17B were selected as representatives of the North Basin. Grey dotted lines indicate the isobaths (in m), and dashed line represents the boundary of the basin. This figure was reproduced from the website of the Geospatial Information Authority of Japan (https://www.gsi.go.jp) and supplemented with latitude and longitude lines. This map is licensed under the Government of Japan Standard Terms of Use (Ver.2.0). The Terms of Use are compatible with the Creative Commons Attribution License 4.0 (CC BY 4.0).

USA). At stations 12B and 17B, water samples were collected with a bucket at 0 m, with 10-L Niskin samplers at 2.5 m, and with 5-L Niskin bottles on a rosette sampler (AWS; JFE Advantech Co. Ltd., Kobe, Japan) at 5, 10, 15 and 20 m after *in situ* FRRf measurement (described

**Table 2. Sampling date, Station ID, light environment ($E_0$ and $K_d$), and time periods for FRRf and $^{13}$C incubation on each sampling date.**

| Date | Station | $E_0$ $\mu$ mol m$^{-2}$ s$^{-1}$ | $K_d$ m$^{-1}$ | FRRf periods | $^{13}$C incubation periods | Light-dark bottle incubation periods |
|---|---|---|---|---|---|---|
| 23 Jul 2018 | 12B | 1630 | 0.433 | 10:30–10:50 | 14:00–17:00 | |
| 30 Jul 2018 | 17B | 705 | 0.328 | 11:10–11:30 | 14:30–17:30 | |
| 28 Aug 2018 | 9B | 650 | 1.727 (0–2 m) | 11:00–11:20 | 15:00–18:00 | 15:00–18:00 |
| | | | 0.971 (2–5 m) | | | |
| 13 Sep 2018 | 12B | 440 | 0.541 (0–6 m) | 10:15–10:35 | 14:30–17:30 | 14:30–17:30 |
| | | | 0.261 (6–20 m) | | | |
| 18 Sep 2018 | 6B | 124 | 1.029 | 9:35–10:00 | 14:15–17:15 | 14:15–17:15 |
| 10 Oct 2018 | 9B | 176 | 1.001 | 10:40–10:50 | 14:45–17:45 | |
| 25 Oct 2018 | 9B | 1090 | 0.881 | 10:00–10:15 | 14:30–17:30 | |
| 16 Nov 2018 | 12B | 901 | 1.198 (0–2 m) | 9:45–10:00 | 14:00–17:00 | |
| | | | 0.375 (2–20 m) | | | |
| 10 Dec 2018 | 6B | 221 | 1.117 | 10:30–10:45 | 14:45–17:45 | |
| 19 Dec 2018 | 17B | 255 | 0.451 (0–6 m) | 11:00–11:30 | 16:00–19:00 | |
| | | | 0.224 (6–20 m) | | | |
| 28 Jan 2019 | 9B | 235 | 1.029 (0–2 m) | 10:30–10:45 | 14:30–17:30 | |
| | | | 0.387 (2–4.5 m) | | | |
| 7 Feb 2019 | 12B | 279 | 0.536 (0–2 m) | 9:35–10:00 | 14:30–17:30 | |
| | | | 0.252 (2–20 m) | | | |
| 19 Apr 019 | 9B | 445 | 1.916 (0–1 m) | 10:40–11:00 | 14:00–17:00 | |
| | | | 0.607 (1–5.5 m) | | | |
| 10 May 2019 | 12B | 1711 | 0.894 (0–2 m) | 10:10–10:30 | 15:00–18:00 | |
| | | | 0.263 (2–20 m) | | | |

$E_0$: sub-surface PAR at 0 m at the time of FRRf sampling; $K_d$: diffuse attenuation coefficient of PAR calculated from Eq (1). Additional calculations of $K_d$ were performed for each layer when the logarithmic slope significantly changed with depth.

below). Likewise, at stations 6B and 9B, water samples were collected with a bucket at 0 m, and with an electric pump at 2 and 3 m (6B) or 4 m (9B).

Macro-nutrients and dissolved inorganic carbon (DIC) concentrations were determined from an aliquot of a 100 mL subsample collected at each depth, and immediately filtered through a syringe-type membrane filter (0.2 μm pore size, Acrodisc syringe filter; Pall Corporation, Ann Arbor, MI, USA) using clean techniques. The filtered samples were stored at −20°C until nitrate, nitrite, ammonia and phosphate analyses using an ion chromatograph system (Dionex Integrion HPIC system; Thermo Scientific, Waltham, MA, USA). DIC was analysed using a total carbon analyzer (TOC-L; Shimadzu, Kyoto, Japan). For chlorophyll *a* (Chl-*a*) analysis, 50–200 mL samples were filtered onto a 25-mm glass-fibre filter (0.7 μm nominal pore size, GF/F; GE Healthcare, UK Inc., Little Chalfont, UK). Chl-*a* was extracted with *N,N*-dimethylformamide for 24 h in the dark [63] and then stored at −80°C. The Chl-*a* concentration was determined with a 10-AU fluorometer (Turner Designs, Sunnyvale, CA, USA).

We measured photosynthetically active radiation (PAR, 400–700 nm) underwater from a depth of 30 m to the surface using a $2\pi$ PAR sensor (CTG Ltd., West Molesey, UK) along with the FRR fluorometer. We determined the diffuse attenuation coefficient $K_d$ (m$^{-1}$) with an exponential function as follows:

$$E_{bot} = E_{top}\exp(-K_d \times dZ) \tag{1}$$

where $E_{bot}$ and $E_{top}$ are incident PAR (μmol photon m$^{-2}$ s$^{-1}$) at the bottom and top of the

sampling layer, respectively, and $dZ$ is the thickness of the layer. When the logarithmic slope of $K_d$ did not change with depth, it was calculated as $E_Z = E_0 \exp(-K_d \times dZ)$, where $E_0$ is PAR at 0 m. When the logarithmic slope of $K_d$ significantly changed with depth, we divided the water column into two layers at the depth where $K_d$ changed and calculated it for each layer (Table 2).

## FRRf measurements and photophysiological parameters

*In situ* induced Chl-*a* fluorescence profiles were measured vertically with a multi-excitation wavelength fast repetition rate fluorometer (FRRf) system (FastOcean, S/N 17-0053-002; CTG Ltd., UK). The field-type FRRf was equipped with two chambers for ambient light and dark readings. To remove ambient light noise, an optical bandpass filter (<670 nm) was attached above the light chamber. The dark chamber has black housing and piping with a pump to ensure that samples are measured under completely dark conditions after 1–2 s of dark adaptation. Each chamber has three light-emitting diodes (LED) providing flash excitation energy centred at 444, 512 and 633 nm (S1A Appendix). Here, 444 nm (blue) corresponds to the absorption peak of Chl-*a*, while 512 nm (green) and 633 nm (orange) correspond to the absorption peaks of phycoerythrins and phycocyanins [49]. We employed four LED combinations to evaluate the green and orange excitation flashes: (1) 444 nm, (2) 444 and 512 nm, (3) 444 and 633 nm, and (4) 444, 512 and 633 nm. We applied a single turnover method, which was consistent with a saturation phase (100 flashlets with 2 μs pitch) and a relaxation phase (40 flashlets with 50 μs pitch). This sequence was repeated eight times with a 100-ms interval for each LED combination. All combinations were repeated at least five times every 5 m from 20 to 10 m and every 2 m from 10 m to the surface at 12B and 17B, and every 0.5 m from the bottom to the surface layer at 6B and 9B during the up-cast, respectively. The power of flashlets ($E_{FRRf}$) and the gain of the extra high tension of the photomultiplier tube (PMT eht) were optimised by FastPro8 software (version 1.0.50; CTG Ltd.). All FRRf measurements were performed between 09:30 and 11:30 (Table 2).

The concentration of PSII reaction centre (RCII, nmol m$^{-3}$) was estimated fluorometrically according to Oxborough et al. [28] as follows:

$$\text{RCII} = K_R / E_{FRRf} \times F_o / \sigma_{PSII} \times 10^{-9} \qquad (2)$$

where $K_R$ is an instrument-specific constant (photons m$^{-3}$ s$^{-1}$), $F_o$ is the fluorescence intensity at the zeroth flashlet of a single turnover measurement when all RCII are open, and $\sigma_{PSII}$ is the absorption cross section of PSII photochemistry in the dark (m$^2$). A recent study showed that $K_R / E_{FRRf}$ can vary among phytoplankton taxonomic groups and growth conditions, and affect the estimation of productivity [64]. In this study, we did not examine sample-specific $K_R / E_{FRRf}$ values. Instead, we used a constant value as in the work of Wei et al. [65], and taxonomic group and nutrients were assessed as the factors affecting $\Phi_{e,C}$ by a statistical model approach (mentioned below).

The RCII-specific rate of electron transport based on FRRf ($J_f$, μmol electrons nmol RCII$^{-1}$ s$^{-1}$) was calculated based on the Sigma Algorithm installed in FastPro8 [28, 36]:

$$J_f = E \times \sigma_{PSII}' \times (1 - C) \qquad (3)$$

where $E$ is the incident PAR at each sampling depth (μmol photon m$^{-2}$ s$^{-1}$), $\sigma_{PSII}'$ is the absorption cross section of PSII photochemistry under ambient light (m$^2$), and $(1 - C)$ is the fraction of RCII in the open state, which is assumed to be $qP$ ($= (F' - F_O')/(F_m' - F_O')$). Thus, the electron transport rate per water volume ($JV_f$, μmol electrons m$^{-3}$ s$^{-1}$) was derived by $J_f \times$ RCII. Special permission is required for *in situ* bottle incubations in Lake Biwa, limiting our

ability to conduct approximate time-dependent comparisons. Thus, we conducted bottle incubation measurements of carbon fixation and oxygen production rates after returning to the laboratory from 14:00 to 18:00 close to the same time of day (Table 2). Previous studies suggested that the diel changes in $J_f$ are relatively small during the daytime [66, 67]. To check the diel changes in $J_f$ of the phytoplankton community in Lake Biwa, we conducted extra measurements of the $J_f$ in the laboratory (at the start of bottle incubation) using the Act2 system (CTG, Inc.) on 28th January, 19th April and 10th May in 2019, and confirmed that the $J_f$ were quite comparable between *in situ* and laboratory conditions (S2 Appendix).

Phytoplankton can dissipate excessive energy as heat in PSII, and after PSII through alternative electron flows (AEFs) such as the Mehler reaction (for reviews, see [33, 68]). Among these, the proportion of heat dissipation relative to the total absorbed energy was estimated as the normalised Stern–Volmer quenching coefficient (NPQ$_{NSV}$), which is equal to $F_O'/F_v'$ [69]. This is commonly used to compare non-photochemical quenching among phytoplankton communities that have different light histories [27, 38, 70]. Finally, the maximum quantum efficiency of PSII was evaluated by $F_v/F_m$ [71]. It should be noted that, according to previous studies, PSII fluorescence of cyanobacteria could be lower under dark conditions due to the lag time associated with the transition from an illuminated to a dark state (State 1 to State 2) expressed in seconds to minutes [72–74]. Thus, when cyanobacteria are dominant in a community, $F_O$ and $F_O'$ could be overestimated [75], and cause underestimation of $qP$ but overestimation of NPQ$_{NSV}$ and RCII.

## Evaluation of excitation wavelength combination

To assess the performance of the four wavelength combinations in the natural phytoplankton communities, we compared the minimum fluorescence yield *Fo* measured at each wavelength combination during cyanobacterial blooms (at 9B on 28th August in 2018), and when diatoms and zygnematophytes dominated (at 12B on 13th September and at 6B on 18th September in 2018). We also compared electron transport rates estimated by FRRf ($JV_f$, µmol e⁻ m⁻³ s⁻¹) and by the light-dark bottle method ($JV_O$, µmol e⁻ m⁻³ s⁻¹) [76] to verify the accuracy of the estimated $JV_f$.

The light-dark bottle method is known to underestimate GPP due to differences in respiration rate between dark and light conditions [77, 78], and oxygen oversaturation [79]. Nevertheless, since the fixation of one molecule of $CO_2$ theoretically requires at least four electrons through photosynthesis, the method is still practical to determine whether or not the $JV_f$ determined by the FRRf was underestimated. Water samples from each layer were poured into two or four 100-mL glass bottles. To measure the community respiration rate, another aliquot of sample water from each layer was poured into two 100-mL dark bottles. All bottles were incubated for 3 h in a growth chamber (HCLP-880PF; Nippon Medical and Chemical Instruments Co., Ltd., Osaka, Japan). Oxygen concentration of each bottle was measured by the optical oxygen spots and probe (Fibox 4; PreSens, Regensburg, Germany) before and after incubation. $JV_O$ was derived as follows:

$$JV_O = (NP_O - R_d) \times 3.47 \times 10^{-2} \tag{4}$$

where $NP_O$ is net oxygen evolution rate (mg $O_2$ m⁻³ h⁻¹), $R_d$ is dark respiration rate (mg $O_2$ m⁻³ h⁻¹), and $3.47 \times 10^{-2}$ is a conversion factor from hours to seconds, from mg $O_2$ to µmol $O_2$, and 4 mol e⁻ for 1 mol $O_2$ evolution. Plots of $JV_O$ against light intensity were fitted in a two-parameter model, as described by Webb et al. [80].

Additionally, quality control of all FRRf data measured with each excitation combination was assessed by the probability of an RCII being closed during the first flashlet of a single

turnover saturation phase under dark ($R\sigma_{PSII}$) and ambient light ($R\sigma_{PSII}'$) by FastPro8 software. Although FastPro8 adjusted $R\sigma_{PSII}$ and $R\sigma_{PSII}'$ around 0.05, these values changed in a manner dependent on depth, light environment, and phytoplankton community composition. Thus, we compared the number of successful observations, $R\sigma_{PSII}$ and $R\sigma_{PSII}'$, among four combinations after rejecting extremely low-quality data ($R\sigma_{PSII}$ or $R\sigma_{PSII}' < 0.03$ or $> 0.08$).

### Phytoplankton identification and enumeration

For enumerating phytoplankton, 50 mL of each sample was fixed with Lugol's solution (1% final concentration). After 24 h of settling in the dark, the supernatant was removed gently and the sample was concentrated to 15 mL. Cells were counted under a light microscope at ×200 magnification where the size and volume of cells for each group were measured using cellSens software (Olympus, Tokyo, Japan) based on the work of Hillebrand et al. [81]. All phytoplankton species were categorized into eight groups: diatoms, cyanophytes, small chlorophytes, zygnematophytes, cryptophytes, chrysophytes, dinoflagellates and euglenophytes. The phytoplankton community composition was assessed based on the carbon biomass converted from the biovolume [82].

### $^{13}$C uptake rate

Gross primary production per water mass ($GP_C$, mg C m$^{-3}$ h$^{-1}$) was determined by a $^{13}$C-based method based on Hama et al. [16]. Water samples from each layer were taken to the laboratory and poured into two or four 500-mL polycarbonate bottles (Nalgene, Rochester, NY, USA), and spiked with NaH$_{13}$CO$_3$ to a final concentration of ca. 10% of ambient total inorganic carbon [83]. Incubations were initiated within 3 h after samples had been collected. Production experiments were conducted in incubators where the temperature and light environment were controlled in a growth chamber along with an oxygen evolution experiment. Incubation PAR levels were set as in Table 3. Incubation PAR intensity was manipulated using black mesh filters covering polycarbonate bottles. Incubation temperature was set to the mean of respective sampling depths. The samples from 12B on 23rd July and 17B on 30th July were incubated under ambient light percentages (described above) and temperature on the balcony of Lake Biwa Environmental Research Institute (Shiga, Japan). Although incubation temperature was not controlled on 23rd and 30th July, it changed <1.5˚C during the incubations.

All incubations were conducted for 3 h. After each incubation period, water samples were filtered through pre-combusted (at 450˚C for 4 h) 25-mm glass-fibre filters (0.7 μm nominal pore size, GF/F). The filters were stored at −20˚C until final analysis. The carbon stable isotope ratio δ$^{13}$C was measured using a Delta V Advantage isotope ratio mass spectrometer coupled with a Conflo IV interface and a Flash 2000 elemental analyzer (Thermo Fisher Scientific, Waltham, MA, USA) at the Isotope Research Institute (Yokohama, Japan). $GP_C$ was calculated according to Hama et al. [16]. To compare with the measured $J_f$ in Eq (3), $GP_C$ was converted

**Table 3. Sampling depth for each incubation PAR intensity.**

| Station | | | Incubation PAR | | |
|---|---|---|---|---|---|
| | **100%** | **65%** | **30%** | **10%** | **1.6%** |
| 12B and 17B | 0 m | 0 m | 2.5 m | 5 m | 10 m |
| 9B | 0 m | 0 m | 2 m | 2 m | 4 m |
| 6B | 0 m | 0 m | 2 m | 2 m | 3 m |

to RCII-specific primary production rate ($PB_C$, mg C nmol $RCII^{-1}$ $h^{-1}$) as follows:

$$PB_C = \frac{GP_C}{RCII} \tag{5}$$

## Spectral correction

To account for the difference in spectral distribution and primary production response between ambient light field and artificial incubator light sources, we applied a spectral correction to the data set to reduce any possible discrepancies between the methods. First, to correct the differences in the spectral distribution of excitation flash of FRRf and ambient light in the water column, $\sigma_{PSII}$ was adjusted by the spectral correction factor (SCF) following previous studies [27, 34]:

$$SCF = \frac{\sum_{700}^{400} a_{phy}^*(\lambda) \times E_{in\ situ}(\lambda) \sum_{700}^{400} E_{FRRf}(\lambda)}{\sum_{700}^{400} a_{phy}^*(\lambda) \times E_{FRRf}(\lambda) \sum_{700}^{400} E_{in\ situ}(\lambda)} \tag{6}$$

where $a_{phy}^*(\lambda)$ is the Chl-$a$ specific absorption spectrum of phytoplankton ($m^2$ mg Chl-$a^{-1}$), and $E_{in\ situ}(\lambda)$ and $E_{FRRf}(\lambda)$ are the spectral distribution of irradiance in the water column and excitation flash of FRRf, respectively. The $a_{phy}^*(\lambda)$ and $E_{in\ situ}(\lambda)$ were estimated from models described in previous studies where the spectral irradiance in the water column was estimated as follows [84, 85]:

$$E_{in\ situ}(\lambda, z) = E_0(\lambda) exp(-[a_W(\lambda) + a_{CDOM}(\lambda) + a_{NAP}(\lambda) + chl \times a_{phy}^*(\lambda)] \times z) \tag{7}$$

where $\lambda$ is wavelength between 400 and 700 nm, $a_W$, $a_{CDOM}$ and $a_{NAP}$ are absorption spectra of pure water, CDOM and non-algal particles, respectively ($m^{-1}$), $chl$ is Chl-$a$ concentration (mg $m^{-3}$) and z is depth (m). $a_{phy}^*(\lambda)$ was estimated for each date and depth by Paavel's model [86] for August, and Ylöstalo's model [87] for other months, in accordance with the species composition of the phytoplankton community (S3 Appendix). In Lake Biwa, *Anabaena affinis* and *Aphanothece* sp. dominated >90% of the cyanobacterial biomass at 0 and 2 m, respectively, in August 2018. The former species has phycocyanin (PC) [49], whereas the latter has PC and phycoerythrin (PE) [88, 89]. Paavel et al. [86] described a summer cyanobacterial bloom that consisted of *Anabaena*, *Aphanizomenon*, and *Gloeotrichia*. The former two species have PC [49], and later species has PC and PE [90]. We tested the $a_{phy}^*(\lambda)$ of pure *Anabaena* [49] culture in August 2018 in Lake Biwa, but the SCF changed to only 3%. We believe that Paavel et al. $a_{phy}^*(\lambda)$ coefficients are better for the abovementioned reason.

The $a_W$ was estimated by a common spectral model determined by Pope and Fry [91] (S4A Appendix). The $a_{CDOM}$ was estimated by the equation $a_{CDOM} = a_{CDOM}(320) exp(-S(\lambda-320))$ [92]. We used previously measured, average values of limnetic sites from each basin, 1.03 and 2.28, for $a_{CDOM}(320)$ for the North Basin and the South Basin, and 0.017 for $S$ [85, 92] (S4B Appendix). $a_{NAP}$ was estimated as $a_{NAP} = a_{NAP}(440) exp(-S(\lambda-440))$ [93]. We used 0.264 for $a_{NAP}(440)$ and 0.004 for $S$ as typical values for the area (S4C Appendix). The calculated $E_{in\ situ}(\lambda, z)$ was adjusted with observed PAR intensity at each depth during each sampling date.

For the spectral distribution of incident sunlight $E_0(\lambda)$ on each sampling date, we referred to the solar radiation spectrum database of central Japan in 2015 [94]. To fit the incident angle of sunlight and the weather conditions on each sampling date, we used spectral data at 10 AM on 30th January, 9th February, 18th April, 17th May, 2nd and 31st July, 26th August, 7th and 19th September, 10th and 26th October, 10th November, and 9th and 19th December corresponding to each sampling date (S5 Appendix). For spectral irradiance during $^{13}$C incubation on 23rd and 30th July, we used spectral data at 4 PM on 2nd and 31st July from the database.

Calculated SCFs are listed in the S1 Table. The SCF for PAR intensity of each growth chamber was also adjusted with the spectral distribution of the light source (S1B Appendix) and $a_{phy}^*(\lambda)$ in the same manner as in Eq (6).

## Derivation of photosynthetic parameters

To calculate $\Phi_{e,C}$ (mol e$^-$ mol C$^{-1}$) from $J_f$ and $PB_C$ assessed at different light levels, photosynthesis versus irradiance curves (*P-E* curves) were obtained by curve-fitting using two *P-E* models. When $J_f$ or $PB_C$ showed photoinhibition (*i.e.* decreasing $J_f$ or $PB_C$ with increasing $E$ after the light-saturated phase), *P-E* curves were fitted in a three-parameter model, as described in earlier studies [95–97]. When there was no photoinhibition, *P-E* curves were fitted in a two-parameter model as described by Webb et al. [80].

It should be noted that the phytoplankton community structures were occasionally different among the layers of the water column, but the relationships between $J_f$ or $PB_C$ and $E$ were well fitted in the photosynthesis models, as in previous studies [25, 30]. All parameters in the fitted models were calculated by function nls() in R ver. 3.4.3 [98]. The $PB_C$ values corresponding to underwater $E$ of *in situ* FRRf measurements were extracted from the *P-E* curves. $PB_C$ can be functioned by $J_f$ with electron requirement for carbon fixation ($\Phi_{e,C}$) as follows:

$$PB_c = J_f/\Phi_{e,c} \times 43.2 \tag{8}$$

where 43.2 is the conversion factor from seconds to hours, and from μmol C to mg C.

## Data transformation and GLM for $\Phi_{e,C}$

Because FRRf measurements were inhibited by high light intensity (typically >1,000 μmol photons m$^{-2}$ sec$^{-1}$), presumably there were sampling biases between shallow and deep layers. To avoid such bias, we subsampled 240 observations from the data set on each sampling date by bootstrap sampling with replacements: 60 observations from four layers above the euphotic zone (0–3.75, 3.75–7.5, 7.5–12.5 and 12.5–17.5 m) for the North Basin and 80 observations from three layers (0–1, 1–3 and 3–5.5 m) for the South Basin. Here a total of 3360 observations (1680 observations for the North Basin and 1680 observations for the South Basin) were used in all analyses of this study.

As in Eq (8), $\Phi_{e,C}$ is defined as $J_f/PBc \times 43.2$. We modelled $\Phi_{e,C}$ using a generalised linear model (GLM) with gamma error distribution and a log-link function using the glm() function in R. We treated $\Phi_{e,C}$ as the dependent variable and environmental factors as the explanatory variables. To avoid collinearity between the explanatory variables, Spearman's $\rho$ between all candidate factors was tested with a significance level of $p < 0.05$, and parameters with $\rho \geq 0.7$ were regarded as collinear variables. We selected water temperature, PAR, turbidity, dissolved oxygen (DO), NH$_4$, NO$_3$ + NO$_2$, PO$_4$, $F_v/F_m$, $\sigma_{PSII}$, Chl-$a$, and the fractions of diatom, cyanobacteria and cryptophytes in the phytoplankton community as the explanatory variables in GLM. Explanatory variables were standardised (mean 0 and standard deviation 1) after log transformation. NH$_4$ concentration and the fraction of phytoplankton group biomass lower than the detection limit were treated as 0.1 μmol L$^{-1}$ and 0.1%, respectively. Multicollinearity of variables was further tested with the variance inflation factor (VIF) [99] using vif() in the package 'car' [100] in R. A set of all possible sub-models was generated with the dredge() function in the package 'MuMIn' [101], and the sub-models were ranked based on the Akaike information criterion (AIC) [102].

## Results

### Evaluation of excitation wavelength combination

Vertical profiles of temperature showed weak stratification at Station 9B on 28th August and at Station 12B on 13th September (Fig 2A and 2D), but not at Station 6B on 18th September (Fig 2G). Chl-*a* concentration reached 42 μg L$^{-1}$ at 2 m at Station 9B due to a cyanobacterial bloom (Fig 2C). Vertical profiles of the minimum PSII fluorescence yield (*Fo*) showed variability among four combinations of excitation wavelengths during cyanobacterial blooms (Fig 2B), but not for communities dominated by diatoms and zygnematophytes (Fig 2E and 2H). For example, *Fo* values derived from excitation light at 444 nm and 444 + 512 nm were lower than those for excitation combinations of 444 + 633 nm and 444 + 512 + 633 nm when cyanobacteria were dominant at depths of 0 and 2 m at Station 9B (South Basin) in August (Fig 2B and 2C). On the other hand, there were no clear differences in *Fo* profiles at Station 12B (North Basin) on 13th September and Station 6B (South Basin) on 18th September when diatoms and zygnematophytes dominated (Fig 2E, 2F, 2H and 2I). Further, the relationship between $JV_f$ and PAR intensity showed the utility of 633 nm for revealing signatures of cyanobacterial photosynthesis (Fig 3). For example, $JV_f$ was clearly lower than $JV_O$ using excitation at 444 nm and 444 + 512 nm, but not at 444 + 633 nm and the combination of all three wavelengths at Station 9B on 28th August during a cyanobacterial bloom (Fig 3A). No clear differences in $JV_f$ were observed between the combinations of wavelengths at Station 12B on 13th September and Station 6B on 18th September 2018 (Fig 3B and 3C).

The data quality of FRRf measurements during the study period showed reasonable stability (Table 4). Upon rejection of low-quality data (*e.g.* with $R\sigma_{PSII}$ or $R\sigma_{PSII}'$ < 0.03 or > 0.08), the number of successful observations was found to be highest when PSII was excited with a combination of three wavelengths, followed by excitation light at 444 + 633 nm, 444 + 512 nm, and 444 nm. Median values of both $R\sigma_{PSII}$ and $R\sigma_{PSII}'$ were also near the optimal value (0.05) when the three wavelengths were combined.

### Development of $\Phi_{e,C}$ model

**Environmental and biological conditions.** Ancillary measurements of water temperature, DO concentration, turbidity, Chl-*a*, $NO_2 + NO_3$, $NH_4$ and $PO_4$ concentrations from each sampling showed clear spatial and seasonal variability (Table 5). Water temperature varied from 7.5 to 30.2°C in the North Basin and 7.5 to 28.5°C in the South Basin throughout the study period (Table 5). $NO_2 + NO_3$ and $NH_4$ concentrations were lower in summer and autumn, and higher in winter at both basins throughout the study period. $PO_4$ did not show clear seasonal changes and was always lower than 0.04 μmol L$^{-1}$ in both basins throughout the study period.

At all sampling dates, diatoms, zygnematophytes, cyanobacteria and cryptophytes were the dominant groups in the phytoplankton biomass (S6 Appendix). Zygnematophytes, mainly composed of *Staurastrum dorsidentiferum*, *S. sebaldi* and *Micrasterias hardyi*, were always found at all stations throughout the study period, except at Station 6B in December. Diatoms were dominant during summer to early autumn and reached 87% in the phytoplankton biomass at Station 6B in September (S6 Appendix). Cryptophytes were present at a relatively low proportion through the study period, except at Station 6B in December. Cyanobacteria were mainly composed of *Anabaena* (*Dolichospermum*) *affinis* and *Aphanothece* sp. and bloomed at Station 9B on 28th August. Small chlorophytes, crysophytes and dinoflagellates always made up less than 20% of the total phytoplankton biomass. Euglenophytes were very rare and accounted for less than 0.5%of the total biomass throughout the study period.

**Spatiotemporal variation and GLM development for $\Phi_{e,C}$.** To develop an optimal electron requirement for the carbon fixation ($\Phi_{e,C}$) model, we used the data set that was obtained

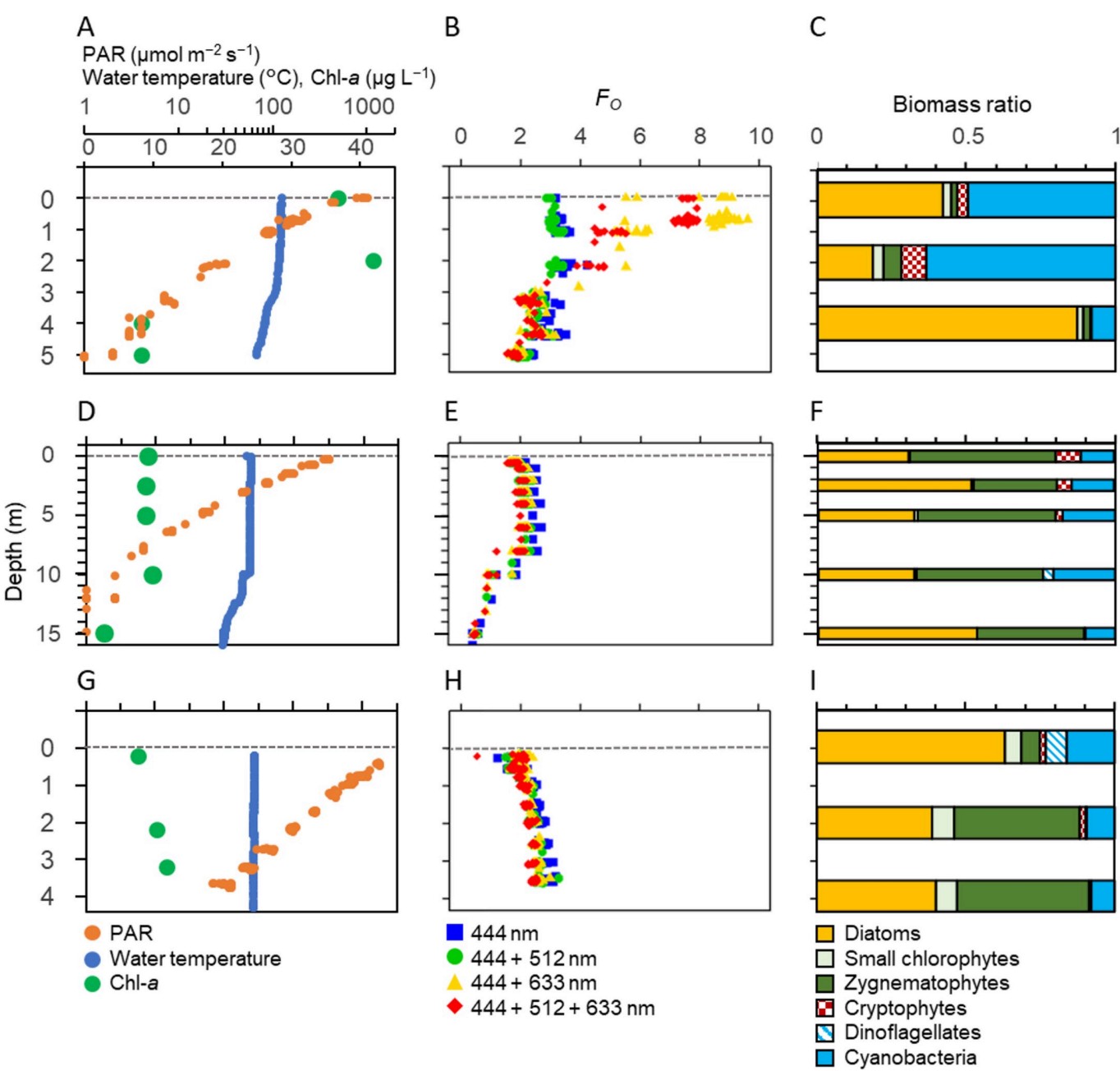

**Fig 2. Comparison of vertical profiles of PAR, water temperature, Chl-*a*, *Fo* and phytoplankton biomass ratios during cyanobacterial blooms and at other times.** Panels showing PAR, water temperature, Chl-*a* (A, D, G), *Fo* (B, E, H) estimated by different combinations of excitation wavelength from the FRRf, and phytoplankton biomass (C, F, I) at Station 9B on 28th August (A, B, C), Station 12B on 13th September (D, E, F) and Station 6B on 18th September 2018 (G, H, I). Grey dashed lines denote 0 m.

by the combination of three excitation wavelengths due to the quality and reliability (Figs 2 and 3). After bootstrap sampling, boxplots of median $\Phi_{e,C}$ were calculated for each sampling date (Fig 4). $\Phi_{e,C}$ changed temporally from 1.1 to 31.0 mol e$^-$ mol C$^{-1}$ and was higher in spring and summer in both the North and South basins. The mean annual $\Phi_{e,C}$ values were 5.6 mol e$^-$ mol C$^{-1}$ for the North Basin, 9.0 mol e$^-$ mol C$^{-1}$ for the South Basin and 7.3 mol e$^-$ mol C$^{-1}$ for all sampling stations.

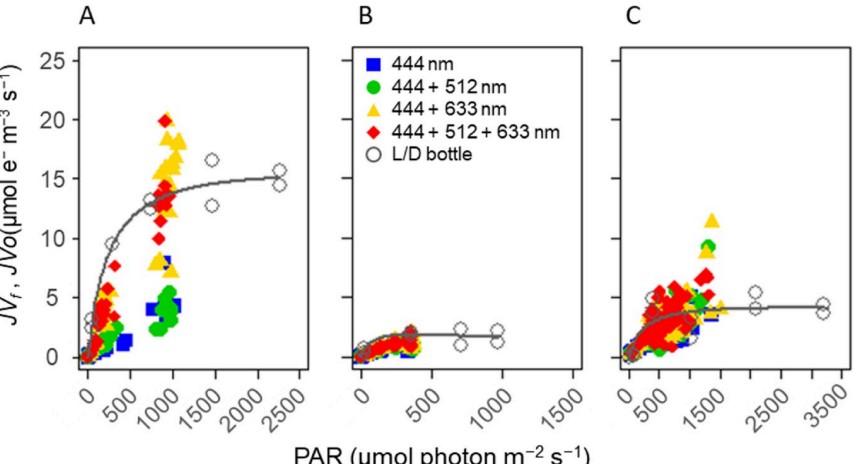

**Fig 3. Scatter plots of $JV_f$ and $JV_O$ during cyanobacterial blooms and at other times.** $JV_f$ was estimated by different combinations of excitation wavelength from the FRRf relative to ambient PAR intensity at (A) Station 9B on 28th August, (B) Station 12B on 13th September and (C) Station 6B on 18th September. The $JV_O$ estimates from the light-dark bottle method are also shown. For $JV_O$, PAR intensity was calculated by the light intensity of growth chambers and SCF (see Materials and methods). The fitted curve is given for $JV_O$ using a two-parameter model [80] to improve visibility.

To select and define the explanatory variables for GLM, we examined correlations between $\Phi_{e,C}$ and all candidate environmental factors (Fig 5). Small chlorophytes, crysophytes and dinoflagellates were excluded because of their low proportions relative to the total phytoplankton biomass. $\Phi_{e,C}$ correlated positively with PAR, temperature, DO, $NPQ_{NSV}$, Chl-$a$, and $\sigma_{PSII}$, and negatively with maximum photochemical efficiency under dark conditions ($F_v/F_m$) throughout the study period. $NPQ_{NSV}$ highly correlated with PAR and $F_v/F_m$ ($\rho = 0.70$ and $-0.96$, respectively). RCII concentration highly correlated with Chl-$a$ ($\rho = 0.70$), and diatoms and zygnematophytes also negatively correlated ($\rho = -0.70$) with each other. Based on the correlation matrix, we selected temperature, PAR, turbidity, DO, $F_v/F_m$, $\sigma_{PSII}$, $NH_4$, $NO_2 + NO_3$, $PO_4$, Chl-$a$, and fractions of diatoms, cyanobacteria, and cryptophytes in the phytoplankton biomass as explanatory variables for the GLM. The explanatory variable 'PAR' may include influences of both PAR and $NPQ_{NSV}$ in the GLM due to the high correlation between the two. Similarly, the explanatory variable 'diatoms' in this analysis may include influences of both diatoms and zygnematophytes.

Among all possible models, the best model with the lowest AIC was the full model without PAR (Table 6). All variables in the best model exhibited VIF < 10 and thus collinearity was negligible. The $R^2$ for the best model was 0.67. Among the explanatory variables, temperature

**Table 4. Comparison of the probability of RCII being closed during the first flashlet of a single turnover saturation phase under dark ($R\sigma_{PSII}$) and ambient light ($R\sigma_{PSII}'$) among four combinations of excitation wavelength.**

| Excitation wavelength | N | $R\sigma_{PSII}$ | | $R\sigma_{PSII}'$ | |
|---|---|---|---|---|---|
| | | Median | Range | Median | Range |
| 444 nm | 1419 | 0.036 | 0.030–0.050 | 0.037 | 0.030–0.075 |
| 444 +512 nm | 1868 | 0.039 | 0.030–0.055 | 0.041 | 0.030–0.071 |
| 444 +633 nm | 1875 | 0.041 | 0.030–0.055 | 0.043 | 0.030–0.070 |
| 444 +512 +633 nm | 1925 | 0.045 | 0.030–0.061 | 0.048 | 0.030–0.078 |

N, number of successful observations.

**Table 5. Physical, chemical and biological (ancillary) conditions on sampling dates.**

| Date | Station | Temperature (°C) | DO (%) | Turbidity (NTU) | Chl-a (μg L$^{-1}$) | NO$_2$+NO$_3$ (μmol L$^{-1}$) | NH$_4$ (μmol L$^{-1}$) | PO$_4$ (μmol L$^{-1}$) |
|------|---------|------------------|--------|-----------------|--------------------|-------------------------------|------------------------|------------------------|
| 23 Jul 2018 | 12B | 17.8–30.2 | 82–116 | 0.3–1.3 | 2.4–13.0 | 0.1–1.1 | 0.2–0.4 | 0.01–0.03 |
| 30 Jul 2018 | 17B | 22.2–27.7 | 87–103 | 0.2–0.9 | 2.4–12.8 | 0.1–0.5 | 0.2–0.2 | 0.01–0.03 |
| 28 Aug 2018 | 9B | 25.0–28.5 | 79–136 | 1.6–8.2 | 8.0–41.9 | 0.4–1.1 | 0.8–1.1 | 0.02–0.02 |
| 13 Sep 2018 | 12B | 19.9–23.7 | 80–100 | 0.4–1.1 | 2.4–9.6 | 1.3–5.9 | 0.2–1.0 | 0.01–0.04 |
| 18 Sep 2018 | 6B | 24.2–24.4 | 102–103 | 2.2–4.3 | 7.6–12.4 | 0.3–1.7 | 1.0–1.2 | 0.01–0.02 |
| 10 Oct 2018 | 9B | 22.1–22.6 | 105–117 | 1.3–2.8 | 14.4–17.0 | 0.4–1.4 | 0.9–1.3 | 0.01–0.02 |
| 25 Oct 2018 | 9B | 19.6–19.8 | 107–110 | 1.0–1.9 | 16.2–17.3 | 0.1–0.4 | 0.8–0.8 | 0.01–0.03 |
| 16 Nov 2018 | 12B | 17.2–17.3 | 96–98 | 0.5–1.0 | 2.4–2.9 | 0.2–0.3 | 0.2–0.3 | 0.01–0.02 |
| 10 Dec 2018 | 6B | 11.5–11.6 | 98–99 | 2.2–2.7 | 7.6–8.0 | 0.1–0.5 | 0.8–0.9 | 0.03–0.03 |
| 19 Dec 2018 | 17B | 12.8–12.8 | 93–97 | 0.1–0.7 | 4.6–5.3 | 3.3–4.2 | 0.5–0.6 | 0.01–0.03 |
| 28 Jan 2019 | 9B | 7.5–8.3 | 96–98 | 1.1–2.6 | 2.5–3.4 | 4.8–5.9 | 1.0–1.2 | 0.02–0.03 |
| 7 Feb 2019 | 12B | 8.9–9.0 | 92–94 | 0.2–0.6 | 4.3–6.7 | 6.3–7.1 | 0.3–0.4 | 0.02–0.03 |
| 19 Apr 019 | 9B | 11.8–13.8 | 105–112 | 0.7–1.5 | 5.9–8.9 | 3.0–3.3 | <LOD | 0.01–0.03 |
| 10 May 2019 | 12B | 12.0–15.7 | 102–118 | 0.3–1.5 | 4.2–12.2 | 1.0–5.3 | <LOD–0.8 | 0.01–0.03 |

The values denote depth ranges of 0–17.5 m for Stations 17B and 12B, 0–4 m for Station 9B, and 0–3 m for Station 6B. NTU, nephelometric turbidity units; LOD, limit of detection.

showed the highest significance in the best model (coefficient of 0.51), followed by cyanobacteria (coefficient of −0.20) and $\sigma_{PSII}$ (coefficient of 0.17). The performances of more parsimonious models were examined to evaluate the laborious sampling effort of nutrients and microscopy analysis of phytoplankton assemblages. The lowest AIC models without nutrients (Model 2), and without nutrients and phytoplankton assemblages (Model 3) were employed.

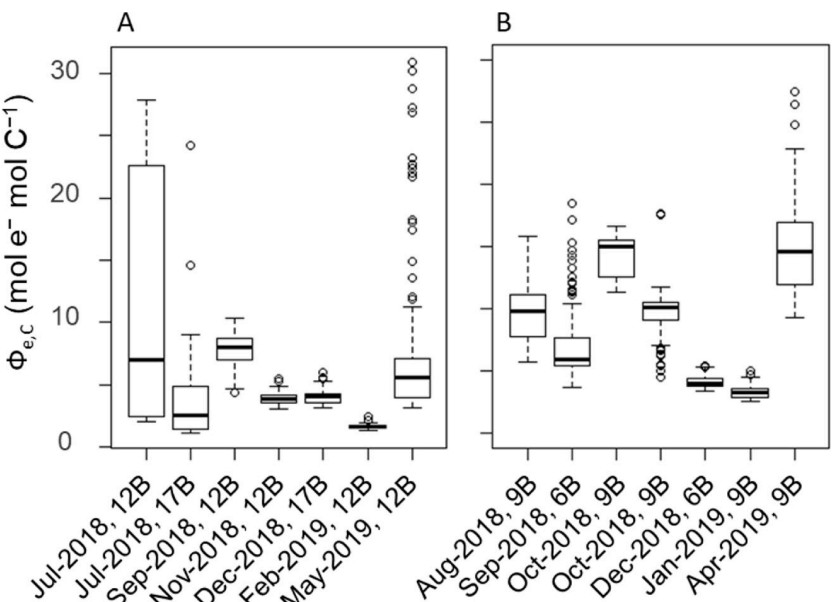

**Fig 4.** Spatial and temporal variability of $\Phi_{e,C}$ of the phytoplankton community in the North Basin (A) and the South Basin (B) throughout the study period. The box plot shows the median (bold line), and 25th (Q1) and 75th (Q3) percentiles. The whiskers indicate 1.5 times the interquartile range (Q3−Q1) below and above Q1 and Q3. Outliers beyond the whiskers were plotted individually. Note: $\Phi_{e,C}$ values were derived from the data measured by the combination of three excitation wavelengths.

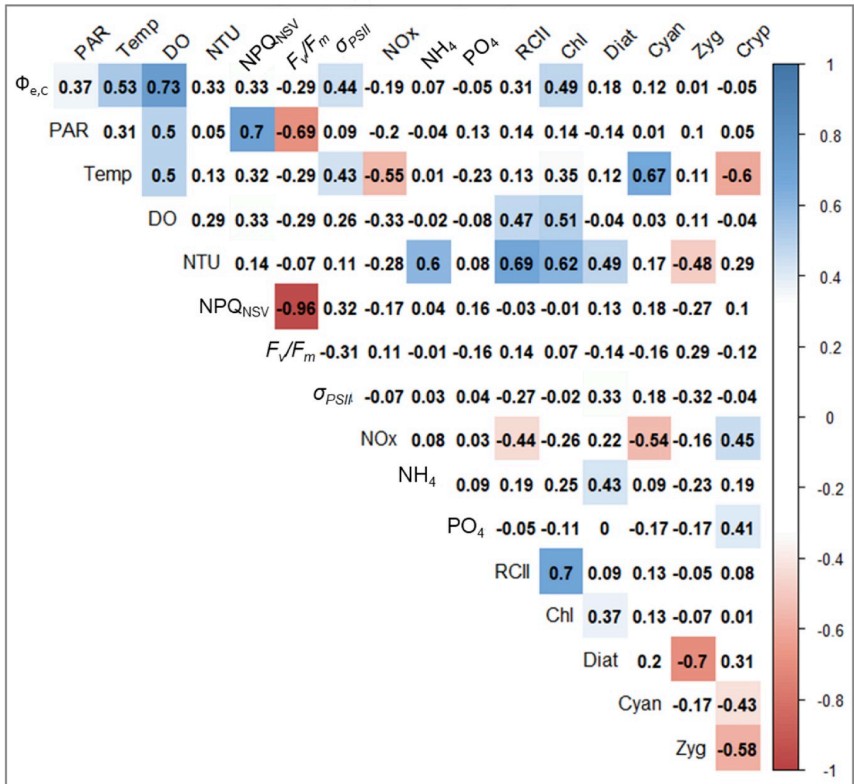

**Fig 5. Matrix of Spearman's $\rho$ between photosynthetic parameters measured by excitation wavelengths of 444 + 512 + 633 nm, physicochemical factors and biomass fraction of each phytoplankton group.** Coloured panels denote statistical significance ($p < 0.05$). Abbreviations of variables are as follows: Temp, water temperature; NTU, turbidity; Chl, Chl-$a$ concentration; Diat, diatom; Cyano, cyanobacteria; Zyg, zygnematophytes; Cryp, cryptophytes.

Model 2 included six variables (temperature, $F_v/F_m$, $\sigma_{PSII}$, cyanobacteria, diatoms and crypto-phytes), while Model 3 included three variables (temperature, $F_v/F_m$ and $\sigma_{PSII}$). The values for $R^2$ for Model 2 and Model 3 were 0.61 and 0.42, respectively. The results of the other sub-models with and without standardisation of variables are listed in the (S2 and S3 Tables).

## Discussion

Recently, studies using FRRf with only one wavelength, around 450 nm, for primary production measurements have been successful [25, 26]. However, the absorption spectrum of phytoplankton is highly dependent on the construction of antenna pigments, and is quite different between cyanobacteria and the other phytoplankton groups [103]. It is well known that the optimal excitation waveband of cyanobacteria is between 550 and 650 nm, while that of eukaryotic algae is between 400 and 550 nm [104, 105]. Therefore, the blue excitation flash at 444 nm can fail to saturate the RCII in cyanobacteria during a single turnover measurement of FRRf [41, 106], and thus underestimate $Fo$ [104] and GPP [21] due to mismatch in wavelength between excitation wavelengths of the FRRf and the absorption spectrum of cyanobacteria. Indeed, the present study demonstrated that $JV_f$ measured with an excitation light of 444 nm (single source) or with excitation lights of 444 + 512 nm was considerably underestimated compared with that in measurements utilising 633 nm, particularly when cyanobacteria dominated (Fig 3). Although we estimated $JV_f$ by each combination of excitation wavelength with SCF, which was calculated from extrapolated $a^*{}_{phy}$, there might be large differences between

**Table 6. Statistical results of the GLM analysis showing the best model (smallest AIC in all models), Model 2 and Model 3 for $\Phi_{e,C}$.**

| Variables | Coefficient | Std. Error | t value | P |
|---|---|---|---|---|
| **Best Model: AIC = 15481, $R^2$ = 0.67** | | | | |
| Intercept | 1.83 | 0.01 | 266.6 | <0.001 |
| Temperature | 0.51 | 0.02 | 25.5 | <0.001 |
| DO | 0.13 | 0.01 | 11.8 | <0.001 |
| Turbidity | 0.10 | 0.01 | 8.0 | <0.001 |
| $F_v/F_m$ | -0.10 | 0.01 | -12.7 | <0.001 |
| $\sigma_{PSII}$ | 0.17 | 0.01 | 17.3 | <0.001 |
| $PO_4$ | -0.02 | 0.01 | -2.1 | 0.034 |
| $NH_4$ | -0.02 | 0.01 | -2.7 | 0.006 |
| $NO_2+NO_3$ | 0.13 | 0.01 | 12.0 | <0.001 |
| Chl-a | 0.10 | 0.01 | 7.0 | <0.001 |
| Cyanobacteria | -0.20 | 0.01 | -16.9 | <0.001 |
| Diatoms | -0.11 | 0.01 | -10.2 | <0.001 |
| Cryptophytes | 0.14 | 0.01 | 10.2 | <0.001 |
| **Model 2: AIC = 15987, $R^2$ = 0.61** | | | | |
| Intercept | 1.84 | 0.01 | 252.2 | <0.001 |
| Temperature | 0.77 | 0.01 | 53.1 | <0.001 |
| $F_v/F_m$ | -0.08 | 0.01 | -10.7 | <0.001 |
| $\sigma_{PSII}$ | 0.10 | 0.01 | 12.1 | <0.001 |
| Cyanobacteria | -0.35 | 0.01 | -32.9 | <0.001 |
| Diatoms | -0.04 | 0.01 | -4.8 | <0.001 |
| Cryptophytes | 0.28 | 0.01 | 25.2 | <0.001 |
| **Model 3: AIC = 17334, $R^2$ = 0.42** | | | | |
| Intercept | 1.89 | 0.01 | 193.2 | <0.001 |
| Temperature | 0.28 | 0.01 | 25.5 | <0.001 |
| $F_v/F_m$ | -0.15 | 0.01 | -14.4 | <0.001 |
| $\sigma_{PSII}$ | 0.22 | 0.01 | 20.4 | <0.001 |

Coefficients were derived for log-transformed and standardised variables (see Materials and methods). AIC and $R^2$ for each model are also shown.

modelled and actual $a^*_{phy}$ around 633 nm, but not around 512 nm in the green part of the spectrum. This is because $a^*_{phy}$ at 630 nm in August, when *Anabaena* spp. dominated, was estimated as 0.008–0.009 m$^{-2}$ mg Chl-$a^{-1}$ (S3 Appendix), while that of cultured *Anabaena* sp. was 0.028 m$^{-2}$ mg Chl-$a^{-1}$ [49]. Further, $a^*_{phy}$ of cyanobacteria can vary with taxonomic group [49] and nutrient availability [107, 108]. Our results suggest that, if the absorption spectrum cannot be measured *in situ*, the excitation light targeting phycobilin antenna pigments should be used to measure $ETR_{PSII}$ in communities where cyanobacteria dominate.

In this study, $\Phi_{e,C}$ in Lake Biwa ranged temporally from 1.1 to 31.0 mol e$^-$ mol C$^{-1}$ (Fig 4). Although our $\Phi_{e,C}$ varied less than that reported in temperate ocean conditions (1.0 to 66.5 mol e$^-$ mol C$^{-1}$ [26]), it was similar to those in an Atlantic Ocean transect (1.1 to 28.2 mol e$^-$ mol C$^{-1}$ [35]) and Ariake Bay (1.2 to 26.6 mol e$^-$ mol C$^{-1}$ [25]), and that in a shallow eutrophic lake (from 14.7 to 38.6 mol e$^-$ mol C$^{-1}$ [60]). Considering that environmental conditions varied substantially among the sampling stations and seasons in this study, the range of our $\Phi_{e,C}$ reasonably represented variations in Lake Biwa, including its oligotrophic (North Basin) and mesotrophic (South Basin) areas.

It should be noted that, because we measured carbon fixation rates during the afternoon, there may have been diurnal variability of phytoplankton productivity. Previous studies

reported that the carbon fixation rate of natural communities is ~1.3 times higher in the morning than in the afternoon because photoinhibition and nutrients depletion suppress community productivity in the afternoon [34, 109]. If we underestimated carbon fixation rates of the communities incubated in the growth chamber, the result might have overestimated $\Phi_{e,C}$. However, $\Phi_{e,C}$ values were paradoxically smaller than theoretical values (4 mol e$^-$ mol C$^{-1}$) during most of the observations (Fig 4), except on 28th August, 13th September, 10th and 25th October in 2018, and 19th April in 2019. Therefore, the influences of $\Phi_{e,C}$ overestimation may be minimal, and we may have actually underestimated $\Phi_{e,C}$ values throughout the study period. The smaller than theoretical $\Phi_{e,C}$ values can be due to underestimation of $JV_f$, especially when cyanobacteria dominated [36, 104], and/or overestimation of carbon fixation rate due to bottle effects [26, 35, 43]. The former case is plausible because the GLM revealed that the effect of cyanobacteria on $\Phi_{e,C}$ was significantly negative (see below). The latter case is more likely in this study, due to mitigation of UV stresses in the growth chamber and an increasing growth rate of cells on bottle walls [110, 111]. It is also possible that the geometric properties of the incubation vessel can alter the light environment on each bottle, and cause calculation errors in ETR$_{PSII}$ and underestimation of $\Phi_{e,C}$ [43]. In any case, although we applied relatively standard methods throughout the study, i.e., FRRf vs. bottle incubations with an isotopic tracer, the results showed $\Phi_{e,C} < 4$, which may reflect core issues in this approach.

The $\Phi_{e,C}$ values $< 4$ have often been observed in laboratory cultures [43, 70], natural communities in open oceans [26, 40, 44], and coastal regions [25, 36, 39]. However, the analysis method for the low $\Phi_{e,C}$ values is not yet unified. Most of the studies included the low $\Phi_{e,C}$ values in the statistical analyses [36, 39, 40, 44, 70], while a few studies did not [25, 26]. In this study, we regarded all of the $\Phi e,C$ values as apparent values as in the previous study [35] since the bottle incubation method can cause artifacts for all bottles. Thus, the low $\Phi e,C$ values were included in the GLM analysis to clarify the apparent effect of environmental and biological factors on the variance of $\Phi e,C$.

The GLM determining $\Phi_{e,C}$ revealed that multiple physicochemical and biological factors, except PAR, were significant in Lake Biwa (Table 6). Previous studies primarily focused on the relationships between light environment, or NPQ$_{NSV}$ and $\Phi_{e,C}$ [25, 27, 34, 38, 40, 65]. The NPQ$_{NSV}$ is phenomenologically correlated with alternative electron flow (AEF) activity, which is activated by excess light and photodamage on PSII [1, 68, 112]. In the present study, PAR intensity was highly correlated with NPQ$_{NSV}$ (Fig 5), and thus we incorporated PAR in GLM for $\Phi_{e,C}$ as a proxy of excess light and NPQ$_{NSV}$. Although PAR was not utilised in the development of the best model, water temperature, DO, $F_v/F_m$, $\sigma_{PSII}$, nutrient conditions and the compositions of the phytoplankton community were selected. Our results suggest that the light environment is not always the primary factor determining $\Phi_{e,C}$; rather, water temperature plays a critical role in the electron requirement for carbon fixation.

Although the effects of temperature on $\Phi e,C$ were not specifically addressed, the results from the present study can be used to discuss the mechanisms related to this relationship. First, increased temperature decreases the $CO_2$ affinity of Rubisco through the acceleration of the $O_2$ evolution rate and reduction of $CO_2$ solubility [113, 114]. $CO_2$ selectivity of Rubisco depends on its form [115]: Form IA Rubisco (cyanobacteria) and form IB Rubisco (cyanobacteria and chlorophytes) have lower selectivity for $CO_2$ than form ID Rubisco (diatoms). In Lake Biwa, cyanobacteria and zygnematophytes were dominant groups during summer and autumn (S6 Appendix). Thus, Rubisco $CO_2$ affinity of the phytoplankton community in Lake Biwa might decrease following the increasing of water temperature and enhance Rubisco oxygenation (photorespiration). Second, to mitigate photorespiration, phytoplankton express $CO_2$ concentration mechanisms (CCMs) [116]. CCMs need ATP or NADPH, which are also needed for driving the Calvin–Benson cycle. Finally, increasing temperature may lead to a

state of chronic photoinhibition through photodamage [117], as well as changes in species composition of the community [40]. Thus, although we did not examine the interaction among all explanatory variables, temperature may have affected $\Phi_{e,C}$ *vis-à-vis* interaction with the other factors such as nutrient stoichiometry [118].

The relationships between multiple environmental factors other than light and $\Phi_{e,C}$ were reported in previous FRRf [35, 36, 40] and PAM studies [119, 120]. Lawrenz et al. [35] investigated the relationships between the $\Phi_{e,C}$ of marine phytoplankton and environmental conditions, and explored the best fitting models with 14 data sets from various geographical regions. They examined water temperature, salinity, optical depth, attenuation coefficient, Chl-*a*, $NO_3$ and $PO_4$ as explanatory variables, and showed that only water temperature, $NO_3$ and $PO_4$ significantly correlated with $\Phi_{e,C}$ in the data set. Their results support our best model, which included water temperature and nutrients. However, Lawrenz et al. also showed that the coefficients and significance of the identified parameters are quite dependent on the region of interest. Thus, future studies are necessary to examine the complex relationship among temperature, light and nutrient conditions in freshwater environments to delineate the specific conditions that explain $\Phi_{e,C}$ discrepancies.

Although three excitation wavelength combinations were used to evaluate the cyanobacterial photosynthesis correctly, the proportion of cyanobacteria in phytoplankton biomass was significant in determining the best $\Phi_{e,C}$ model (Table 6). The proportion of cyanobacteria within the community may cause differences in the light absorption characteristics between cyanobacteria and the other phytoplankton groups; cyanobacteria absorb only 25%–30% of light in PSII, while other phytoplankton absorb 48%–58% [42]. Kromkamp et al. (2008) suggested that most of the Chl-*a* in cyanobacteria is associated with PSI rather than PSII, and detection-limited by fluorometry. Another explanation is underestimation of the RCII concentration under illuminated conditions (Eq 2). This can be caused by a decrease of $F_O$ under dark conditions due to changes in the fraction of excitation energy distribution between PSI and PSII (state transition) [75]. The FRRf method primarily targets pigments associated with PSII; thus, the electron flow initiated by the excited PSI is difficult to evaluate. Johnsen et al. [103] suggested that the fraction of light absorption by PSII, or that of cellular Chl-*a* in PSII, can be a useful proxy as action spectra, or alternative to absorption spectra, to correct the effect of photoprotective carotenoids and PSI-PSII absorption variability. Considering the large differences in regulation of the distribution of light-harvesting pigments and excitation energy between cyanobacteria and eukaryotic algae, the action spectra proxies should provide more accurate measurement of GPP in future study. Further, evaluating the proportion of cyanobacteria in phytoplankton biomass will be crucial for correcting GPP estimation with $ETR_{PSII}$ in natural phytoplankton communities.

As expected, $PO_4$ concentration negatively affected $\Phi_{e,C}$ (Table 6). This result was consistent with previous $PO_4$ manipulation experiments conducted with marine phytoplankton [121]. The effect of $PO_4$ was, however, one of the lowest among all factors (Table 6). The low significance is likely due to the fact that the $PO_4$ concentrations did not change drastically and were generally low (0.01–0.04 µmol $L^{-1}$, Table 5) throughout the study period.

To verify the performance of the $\Phi_{e,C}$ models, we subsequently compared daily GPP estimated with FRRf ($GPP_f$) and $^{13}C$ ($GPP_{13C}$) for the North and South basins on each sampling date. The RCII-specific primary production based on FRRf ($PB_f$, mg C mmol RCII $^{-1}$ $h^{-1}$) was calculated from the $J_f$, and computed $\Phi_{e,C}$ with the best model, Model 2 and Model 3. The relationships between $PB_f$ and PAR were fitted in *P-E* curves with models [80, 95–97], as described in the materials and methods. Finally, daily $GPP_f$ and $GPP_{13C}$ were estimated as follows:

$$GPP = \int_0^Z [RCII_Z \times \sum_{t=1}^L PBi(Z,t)] \tag{9}$$

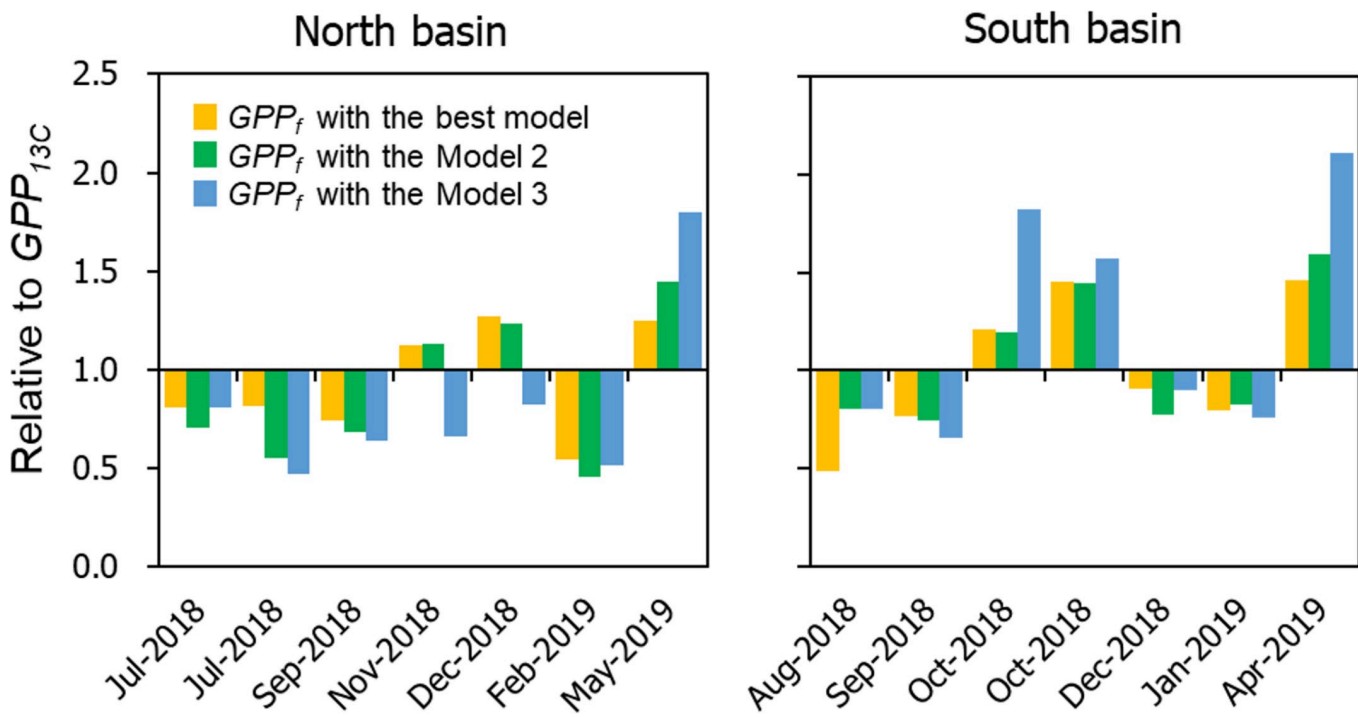

**Fig 6. Relative $GPP_f$ estimated by $ETR_{PSII}$ with the $\Phi_{e,C}$ models against the $GPP_{13C}$ on each sampling date for North and South basins.**

where $RCII_Z$ is the RCII concentration at depth $Z$, $L$ is the day length (h), and $PBi(Z,t)$ is $PB_f$ or $PB_{13C}$ at depth $Z$ and time $t$ (h). The $RCII_Z$ and $PBi(Z,t)$ were calculated every 1.25 m for the North Basin and every 0.5 m for the South Basin based on average values of the observed data. Day length and daily PAR data were obtained from the Japan Meteorological Agency for the North Basin, and were measured by a PAR sensor (PAR-02D; Prede Co., Ltd., Tokyo, Japan) at Otsu for the South Basin.

The estimated $GPP_{13C}$ varied from 71 to 787 g C m$^{-2}$ d$^{-1}$, while $GPP_f$ with the best model, Model 2 and Model 3 varied from 86 to 630 g C m$^{-2}$ d$^{-1}$, 85 to 729 g C m$^{-2}$ d$^{-1}$ and 117 to 906 g C m$^{-2}$ d$^{-1}$, respectively (for date-specific values, see S4 Table). The $GPP_f$ using the best model relative to $GPP_{13C}$ varied from 0.48 to 1.46, suggesting that FRRf parameters with the best $\Phi_{e,C}$ model can reasonably reproduce $GPP_{13C}$ in Lake Biwa (Fig 6). Relative $GPP_f$ with Model 2 and Model 3, including fewer variables, also replicated $GPP_{13C}$ well, varying from 0.46 to 1.59 and 0.47 to 2.11, respectively. The comparative results suggest that, even if nutrients and phytoplankton biomass are not considered, $GPP_{13C}$ can be estimated using the FRRf parameters coupled with temperature for both oligotrophic and mesotrophic areas in Lake Biwa. This model parameterisation is significant for future applications in freshwater ecosystems where environmental conditions and phytoplankton communities can vary spatially and temporally.

In conclusion, the FRRf equipped with an excitation wavelength of 633 nm is effective in estimating $ETR_{PSII}$ for freshwater phytoplankton communities during cyanobacterial blooms. Contrary to our hypothesis, water temperature was the most important determinant factor, while phosphorus concentration was less effective in the $\Phi_{e,C}$ model in Lake Biwa. Further, $GPP_{13C}$ dynamics were effectively estimated from the $ETR_{PSII}$ using the best $\Phi_{e,C}$ model for both oligotrophic and mesotrophic areas in Lake Biwa. The parsimonious model, including only temperature and two photosynthetic parameters, also sufficiently reproduced $GPP_{13C}$.

These results must be interpreted with caution because although $\Phi_{e,C}$ should be mechanistically $\geq 4$ mol e$^-$ mol C$^{-1}$, our study showed $\Phi_{e,C} < 4$ in 9 out of 14 observations. In this study, we define the observed $\Phi_{e,C}$ as apparent values that are included artifacts from methodological issues, such as bottle effects, which can be minimized by a relatively shorter incubation time (~2 h) [122]. Moreover, studies are needed to develop the incubation-free techniques, such as cellular NADP(H) measurement [123]. In this study, the phytoplankton community was verified through microscopy, but fluorometric characteristics from the FRRf may also allow researchers to determine phytoplankton assemblages at high *in situ* spatial and temporal resolution [124], and thus simplify the GPP measurements during cyanobacterial blooms. This study provides strong validation for measurements of primary productivity by FRRf in lakes with large spatiotemporal variabilities of phytoplankton assemblages and environmental conditions from oligo- to mesotrophic lake (or lacustrine) environments. Further, a recent study reported that net primary productivity can also be measured by FRRf when fluorescence induction and relaxation (FIRe) techniques are applied [125]. This suggests that the application of FRRf enables real-time measurements of the PSII photochemical characteristics and the carbon metabolism of phytoplankton simultaneously. In the future, bio-optical measurements will allow researchers to disentangle the causality between anthropogenic nutrient control and fish catch [13, 61, 126], and between climate change and the production of higher trophic levels [7, 127].

## Supporting information

**S1 Table. Spectral correction factor (SFC) for *JV_f* estimation on each sampling date.**
(DOCX)

**S2 Table. All GLM sub-models with standardisation of variables.**
(CSV)

**S3 Table. All GLM sub-models without standardisation of variables.**
(CSV)

**S4 Table. GPP (g C m−2 d−1) estimated by 13C and FRRf with Φe,C models on each sampling date.**
(DOCX)

**S1 Appendix.** Spectral distribution of (A) excitation flash of FastOcean (B) light source of growth chamber.
(PDF)

**S2 Appendix. Scatter plots of *J_f in situ* in the morning, and in the laboratory in the afternoon (at the start of the incubation experiments).** The combination of three excitation wavelengths was used. For $J_f$, PAR intensity was corrected by white background light intensity of the Act2 system and SCF (see Materials and methods). The fitted curve is given by a two-parameter model [80].
(PDF)

**S3 Appendix. Modelled absorption spectra for various Chl-*a* concentrations.** The spectra were calculated with Paavel's model [86] for 30 and 40 μg L$^{-1}$ in August, and Ylöstalo's model [87] for the others.
(PDF)

**S4 Appendix.** Modelled absorption spectra of (A) pure water, (B) CDOM and (C) non-algal particles.
(PDF)

**S5 Appendix.** Spectral distribution of incident sunlight at 10:00 in (A) April to September, (B) October to February and (C) at 4 PM in July in 2015. Each spectrum distribution was referenced to calculate the spectral correction on each sampling date as in Eq (7).
(PDF)

**S6 Appendix. Relative contribution to total phytoplankton biomass by algal groups on each sampling date.**
(PDF)

# Acknowledgments

We would like to thank Dr. Takamaru Nagata in LBERI and Dr. Hiroki Haga in Lake Biwa Museum for sharing their ship time and valuable support during sampling. We sincerely thank Dr. Satoshi Nakada and Maho Iwaki for providing PAR profile data, and Hirokazu Teraishi for assisting with the chemical analysis. The authors would like to thank Enago (www. enago.jp) for the English language review.

# Author Contributions

**Conceptualization:** Takehiro Kazama, Kazuhiro Komatsu.

**Formal analysis:** Takehiro Kazama.

**Investigation:** Takehiro Kazama, Kazuhide Hayakawa, Koichi Shimotori.

**Resources:** Kazuhide Hayakawa.

**Supervision:** Akio Imai.

**Writing – original draft:** Takehiro Kazama.

**Writing – review & editing:** Kazuhide Hayakawa, Victor S. Kuwahara, Koichi Shimotori, Akio Imai, Kazuhiro Komatsu.

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
