## [Decision Letter · Decision Letter 0]

14 Sep 2020

PONE-D-20-24363

Development of photosynthetic carbon fixation model using multi-excitation wavelength fast repetition rate fluorometry in Lake Biwa

PLOS ONE

Dear Dr. Kazama,

Thank you for submitting your manuscript to PLOS ONE. After careful consideration, we feel that it has merit but does not fully meet PLOS ONE’s publication criteria as it currently stands. Therefore, we invite you to submit a revised version of the manuscript that addresses the points raised during the review process.

Overall, the reviewer’ 'opinion about your work is quite positive but they do raise a significant amount of issues that need to be addressed before accepting the paper for publication. Personally, I share the worry about the possibility of a diurnal pattern affecting your results, as mentioned by reviewer 1, so please pay particular attention to that point.

We look forward to receiving your revised manuscript.

Kind regards,

Bruno Jesus, Ph.D

Academic Editor

PLOS ONE

Journal Requirements:

3.We note that [Figure(s) 1] in your submission contain [map/satellite] images which may be copyrighted. All PLOS content is published under the Creative Commons Attribution License (CC BY 4.0), which means that the manuscript, images, and Supporting Information files will be freely available online, and any third party is permitted to access, download, copy, distribute, and use these materials in any way, even commercially, with proper attribution. For these reasons, we cannot publish previously copyrighted maps or satellite images created using proprietary data, such as Google software (Google Maps, Street View, and Earth). For more information, see our copyright guidelines: http://journals.plos.org/plosone/s/licenses-and-copyright.

1.    You may seek permission from the original copyright holder of Figure(s) [1] to publish the content specifically under the CC BY 4.0 license. 

4. Please include a copy of Table 6 which you refer to in your text on page 39.

Reviewers' comments:

Reviewer's Responses to Questions

**Comments to the Author**

1. Is the manuscript technically sound, and do the data support the conclusions?

Reviewer #1: Partly

Reviewer #2: Yes

Reviewer #3: Yes

2. Has the statistical analysis been performed appropriately and rigorously? 

Reviewer #1: I Don't Know

Reviewer #2: Yes

Reviewer #3: Yes

3. Have the authors made all data underlying the findings in their manuscript fully available?

Reviewer #1: No

Reviewer #2: Yes

Reviewer #3: Yes

4. Is the manuscript presented in an intelligible fashion and written in standard English?

Reviewer #1: No

Reviewer #2: Yes

Reviewer #3: Yes

5. Review Comments to the Author

Reviewer #1: The authors used a multi-excitation FRRf to pursue the important topic of connecting (fast, relatively easy) FRRf measures to (slow, expensive) C uptake measures in a well studied model lake.

Lake Biwa has a strong cyanobacterial contribution to the phytoplankton community, which complicates the analyses of FRRf signals.

The work is carefully described and quite extensive.

There is an issue with the design, in that there is a widely known pattern of mid-day/early afternoon depression of photosynthesis, but the study compares morning FRRf measures to afternoon Carbon measures or light dark bottle measures.

Can the authors present data from Lake Biwa to support comparison of morning FRRf to afternoon C metrics?

Abstract:

"Also, the FRRf is still relatively novel,"

developed ~25 years ago, with the classic foundational papers appearing 1998...not really novel.

"The range of Ф e,C in the phytoplankton community varied from 1.1 to 31.0 mol e − mol C −1 during the study period"

Rather:

"The apparent range of Ф e,C in the phytoplankton community varied from 1.1 to 31.0 mol e − mol C −1 during the study period"

The actual ratio of e- : C has to be >= 4

References:

There is a newer paper by Max Gorbunov appearing in MEPS that is highly relevant.

Figures:

Fine.

P.6

"Minimum PSII Fluorescence yield (under background light)"

Rather:

"Minimum PSII Fluorescence yield (under acclimation to background light)"

FO' is measured in darkness or back calculated to estimate fluorescence in darkness with C = 1

P7

"Maximum photochemical efficiency under dark conditions"

No.

For cyanobacteria Fv/Fm is not an accurate proxy for Maximum photochemical efficiency under dark conditions", because of state transitions and non-PSII contributions to measured fluorescence.

Given the topic of the manuscript, using higher plant assumptions/short hand is not wise.

P.8

mg vs. g vs. umol

Why not express everything on a umol basis, instead of flipping units back and forth?

It is a trivial issue, but raises a barrier for estimates like phi eC

p.8:

"However, FRRf tended to over- and under-estimate GPP compared to the 14C and 13C

73 methods, and the light-dark bottle and 18O methods, respectively [32]"

this sentence is unclear, and 'respectively' is almost always a bad idea.

Just list the pairwise comparisons.

P.9

"derived ETRPSII to the GPP rate"

Actually, the relevant comparison is JVPSII to GPP.

ETRPSII does not account for [RCII].

Same comment at line 88, a direct comparison of ETRPSII to GPP does not really make sense, because it does not account for [RCII], or more generally for biomass.

Table 2:

There is a widely known pattern of mid-day/early afternoon depression of photosynthesis, but the study compares morning FRRf measures to afternoon Carbon measures or light dark bottle measures.

Materials & Methods:

"When the

163 logarithmic slope of Kd significantly changed with depth, we calculated it for each layer (Table 2)"

I do not understand this sentence, given the definition of Kd and the preceding equation.

More explanation needed; was E0 reset to the level reaching the top of each depth layer?

"The dark chamber has black housing and piping with a pump to

171 ensure that samples are measured under complete dark after 1–2 s of dark adaptation"

1-2 s is enough time for many cyanos to go from State I (illuminated) to State II (dark).

This can affect the estimate:

qP (=(F ´−FO ´)/(Fm ´−FO ´)

because F' is measured at State I and FO' is measured at State II (lower).

Also, NPQNSV reflects different mechanisms in cyanos and in eukaryotes

Line 213 and elsewhere:

There are minor grammar errors scattered through the text, and I apologize I am too pressed to note them all.

Line 215: as noted above, back to umol e-, why mess around with g & mg O2 etc., Just use umol.

Line 227, back to mg O2... unnecessary flips, confusing conversion factors...

Line 237 etc.

"saturation phase under dark (RσPSII) and ambient light (RσPSII′)"

If you reject every measurement where RsigmaPSII' at the zeroth flashlet is > 0.08, you are going to reject many (most?) of the measures taken under illumination, since even low illumination will close 5-10% of RCII.

Lines 256-260 would be better replaced by a table

Line 323

Line 333 "As in Eq (8), Фe,C is defined as Jf /PBc × 43.2. "

another unnecessary and confusing unit conversion

"Explanatory variables were standardized (mean 0 and standard

341 deviation 1) after log-transformation."

Hmmm.

The dynamic range vs. SD of the different metrics would be quite different; is it reasonable to force them all to the same scale?

Lines 350-371

This passage is highly phenomenological, without reference to the well understand patterns of light capture by cyanobacteria vs. other taxa.

Cyanos have a very small chl bed serving PSII, so very small sigmaPSII445nm compared to eukaryotes with much more chl serving PSII.

This is all well documented, example, Simis et al. Photosynthesis Research

Lines 413-415:

" we used the

414 data set that was obtained by the combination of three excitation wavelengths due to the quality and

415 reliability (Fig 2, 3). "

But, given the changes in cyano dominance, would it better to apply a spectral regime weighted to the community composition?

Line 428 etc.

Before applying an explanatory GLM, would it be wise to screen (or correct) the estimates of phi eC that are lower than theoretically possible?

Explaining measures that must be wrong seems odd.

Line 438:

"The effects of PAR and diatoms in

438 this analysis may have included those of NPQNSV and zygnematphytes."

This sentence does not make sense as written.

Line 482:

"In this study, Фe,C in Lake Biwa ranged temporally from 1.1 to 31.0 mol e− mol C−1 (Fig. 4)."

Rather:

In this study, apparent Фe,C in Lake Biwa ranged temporally from 1.1 to 31.0 mol e− mol C−1 (Fig. 4)."

Фe,C cannot actually go below 4, so measures <4 are telling us about issues with approach.

Line 491:

"The NPQNSV is mechanistically

492 linked with alternative electron flow (AEF) activity, which is activated by excess light and

493 photodamage on PSII [1,64,97]. "

Rather:

"The NPQNSV is phenomenologically correlated

492 linked with alternative electron flow (AEF) activity, which is activated by excess light and

493 photodamage on PSII [1,64,97]. "

Both NPQ and AEF are induced under conditions of excess light; AEF can then loop back to induce NPQ.

Reviewer #2: This is a technically well executed, and well-interpreted, manuscript.

A few points need attention.

Line 54. ‘rapidly adapt’: does this mean adapt in the strict (and preferable) sense of genetic change plus natural selection, or in the more general sense of genetic adaptation plus phenotypic acclimation. The citation of Reynolds (1976) post-dates the initial paper of Collins and Bell (1974; Nature 431, 566-569) on experimental evolution of phytoplankton giving an estimate of the time over which genetic adaptation can occur.

Lines 79-81. Is the increased light availability in the open ocean than coastal waters a result of increased phytoplankton per unit volume in coastal waters, higher concentrations of gelbstoff, more suspended non-living particles, or all three? If it is primarily phytoplankton density, is the total light available in the euphotic zone (less deep in the coastal ocean) less than in coastal waters?

Line 82. Admittedly ‘such as’ is used for the Mehler reaction, into would be useful to mention flavodi-iron proteins and PTOX.

Lines 92-94. Good point.

Lines 104-105. Is ATP availability the main effect of P limitation. What about decreased levels of the three main forms of RNA (rRNA, mRNA, tRNA) that restrict the rate of protein synthesis?

Line 110, 117. Is this P in the water column, or in cells.

Line 151. No method is given for measuring ammonium, but ammonium concentrations are mentioned on (e.g.) line 392.

Line 246. ‘chrysophytes’, not ‘crysophytes’. Also, do the chrysophytes include the Synurophyceae as well as the Chrysophyceae”

Line 352. ‘to a cyanobacterial’

Lines 498-500. Clarify the effects of temperature and oxygen on Rubisco kinetic properties (Galmes et al. 2014 Photosynthesis Research 123: 183-201). Also, while low fluxes through glycolate metabolism is essential for cyanobacteria (Eisenhut et al. 2008 PNAS 106: 17199-17204), carbon concentrating mechanisms restrict Rubisco oxygenase activity and hence photorespiration in air-equilibrium solutions for cyanobacteria and many eukaryotic microalgae (Raven et al. 2017 Journal of Experimental Botany 68: 3701-3716), and even at the increased carbon dioxide concentrations in some freshwaters as a result of carbon dioxide from soil respiration in water entering the lake, as well mineralisation of organic matter entering lakes from the catchment.

Reviewer #3: This is a very thorough investigation of alternative methods for measuring lake production. A high level of detail is provided, and it should be possible for other groups to reference this work and use the same comparison methods in other lakes. The results add to the growing body of evidence which is showing that the relationship between the quantum yields of electron transport and that for carbon fixation is highly variable, but that a good proportion of the variability can be explained by the growth conditions of the phytoplankton. Even more encouraging from a remote sensing point of view, is that simple to measure variables such as lake temperature are weighted highly in the model outcomes.

I have one caveat before publication, however, and it needs consideration. A large set of biooptical measurements are presented here, but a critical variable is missing, and that is the phytoplankton absorption spectrum. Here, the spectrum is modelled rather than measured using Paavel 2016. This could bring a large uncertainty into the results, if the cyanobacterial-domitaed community of Lake Biwa is different from the lakes used by Paaval. This is one point to be discussed - are the modelled absorption spectra realistic?

A second point is a broader critique on the use of absorption for modelling the photosynthetic response. In groups such as cyanobacteria and red algae, there is very little photosynthetic oxygen evolution driven by the blue end of the spectrum. This is due to the completely different antenna of PSII and PSI in these groups. Most of the chlorophyll, and hence blue absorption is associated with PSI. For this reason, it is more accurate to use the action spectrum instead of the absorption spectrum, and this point is made many times in the literature (e.g. Johnsen et al 2007, reference 92). Not many labs are able to measure the PS2 action spectrum, and this (correct) method is not so widely used compared to the default of using absorption. The difference is not so great for diatom-dominated waters, as absorption and action cpectra are rather similar, but there are large differences for cyanos - both marine and freshwater. As the owner of a blue-light only FastTracka myself, I am very aware of this problem. I would like to see a couple of sentences in the discussion mentioning that action spectra could be a better alternative.

I have not made a detailed file of minor corrections on the ms. due to lack of time, but the pdf attached has a few changes.

6. PLOS authors have the option to publish the peer review history of their article (what does this mean?). If published, this will include your full peer review and any attached files.

Reviewer #1: **Yes: **Douglas A. Campbell

Reviewer #2: No

Reviewer #3: **Yes: **Rodney Forster, Hull Marine Lab, University of Hull

---

## [Author Response · Author response to Decision Letter 0]

9 Nov 2020

Editor comments

1. Please ensure that your manuscript meets PLOS ONE's style requirements, including those for file naming. The PLOS ONE style templates can be founda at

>>>We checked and fixed our MS to meet PLOS ONE’s style.

>>>No permits were required for navigation on Lake Biwa. We added this sentence in Methods section. (L127-129)

3.We note that [Figure(s) 1] in your submission contain [map/satellite] images which may be copyrighted. All PLOS content is published under the Creative Commons Attribution License (CC BY 4.0), which means that the manuscript, images, and Supporting Information files will be freely available online, and any third party is permitted to access, download, copy, distribute, and use these materials in any way, even commercially, with proper attribution. For these reasons, we cannot publish previously copyrighted maps or satellite images created using proprietary data, such as Google software (Google Maps, Street View, and Earth). For more information, see our copyright guidelines: http://journals.plos.org/plosone/s/licenses-and-copyright.

>>>The Figure 1 was reproduced from the Geospatial Information Authority of Japan website (https://www.gsi.go.jp) and added latitude and longitude lines. This map is licensed under the Government of Japan Standard Terms of Use (Ver.2.0). The Terms of Use are compatible with the Creative Commons Attribution License 4.0 (CC BY 4.0). We added this explanation in the caption of Fig. 1.

4. Please include a copy of Table 6 which you refer to in your text on page 39.

>>>This was typo of “Table 5”, which is Table 6 in revised version.

Reviewer #1: The authors used a multi-excitation FRRf to pursue the important topic of connecting (fast, relatively easy) FRRf measures to (slow, expensive) C uptake measures in a well studied model lake.

Lake Biwa has a strong cyanobacterial contribution to the phytoplankton community, which complicates the analyses of FRRf signals.

The work is carefully described and quite extensive.

There is an issue with the design, in that there is a widely known pattern of mid-day/early afternoon depression of photosynthesis, but the study compares morning FRRf measures to afternoon Carbon measures or light dark bottle measures.

Can the authors present data from Lake Biwa to support comparison of morning FRRf to afternoon C metrics?

>>> Thank you for the comment. We are aware of the changes in photosynthetic activity during the course of the day, and added explanation to clarify our experimental design (L229-237) as follows,

“Special permission is required for in situ bottle incubations in Lake Biwa, limiting our ability to conduct approximate time-dependent comparisons. Thus, we conducted bottle incubation measurements of carbon fixation and oxygen production rates after returning to the laboratory from 14:00 to 18:00 close to the same time of day (Table 2). Previous studies suggested that the diel changes in Jf are relatively small during the daytime [65,66]. To check the diel changes in Jf of the phytoplankton community in Lake Biwa, we conducted extra measurements of the Jf in the laboratory (at the start of bottle incubation) using the Act2 system (CTG, Inc.) on 28th January, 19th April and 10th May in 2019, and confirmed that the Jf were quite comparable between in situ and laboratory conditions (S2 Appendix)” 

Cosgrove, J., N. R. Moheimani, and M. A. Borowitzka. 2015. Diurnal patterns in phytoplankton photosynthesis, Fremantle Harbour. Journal of the Royal Society of Western Australia 98: 19–27.

Longstaff, B. J., T. Kildea, J. W. Runcie, and others. 2002. An in situ study of photosynthetic oxygen exchange and electron transport rate in the marine macroalga Ulva lactuca (Chlorophyta). Photosynthesis Research 74: 281–293. doi:10.1023/A:1021279627409

Schuback, N., M. Flecken, M. T. Maldonado, and P. D. Tortell. 2016. Diurnal variation in the coupling of photosynthetic electron transport and carbon fixation in iron-limited phytoplankton in the NE subarctic Pacific. Biogeosciences 13: 1019–1035. doi:10.5194/bg-13-1019-2016

Zhao, Y., and A. Quigg. 2015. Study of photosynthetic productivity in the Northern Gulf of Mexico: Importance of diel cycles and light penetration. Continental Shelf Research 102: 33–46. doi:10.1016/j.csr.2015.04.014

 S2 Appendix. Scatter plots of Jf in in situ in morning, and in laboratory in afternoon (at the start of the incubation experiments). The combinations of three excitation wavelength was used. For Jf, PAR intensity was corrected by white background light intensity of Act2 system and SCF (see Materials and methods). The fitted curve is given by a two-parameter model [72].

>>> On the other hand, as it pointed out, previous studies reported that the carbon fixation rate of natural communities is ~1.3 times higher in morning than in afternoon (Zhao and Quigg 2015; Schuback et al. 2016). We further discussed this possibility and its influences on Фe,C estimation during the study period, in the Discussion section (L551-561) as follows, 

“It should be noted that, because we measured carbon fixation rates during the afternoon, there may have been diurnal variability of phytoplankton productivity. Previous studies reported that the carbon fixation rate of natural communities is ~1.3 times higher in the morning than in the afternoon (Zhao and Quigg 2015; Schuback et al. 2016). However, if we underestimated carbon fixation rates (and then overestimated Фe,C) of the communities incubated in the growth chamber, the diel variation of the community photosynthesis rate should be smaller than that in natural conditions. Suppression of production rates can be caused by UV radiation related stress (Franklin et al. 2003), which was negligible in the growth chamber (S1 Appendix B). Furthermore, Фe,C values smaller than theoretical values (4 mol e− mol C−1) were observed in most of the sampling (Fig. 4), except 28th August, 13th September, 10th and 25th October in 2018, and 19th April in 2019. Therefore, the influences of Фe,C overestimation may be minimal, and we may have actually underestimated Фe,C values throughout the study period.”

Franklin, L. A., C. B. Osmond, and A. W. D. Larkum. 2003. Photoinhibition, UV-B and Algal Photosynthesis, p. 351–384. In A.W.D. Larkum, S.E. Douglas, and J.A. Raven [eds.], Photosynthesis in Algae. Springer Netherlands.

Schuback, N., M. Flecken, M. T. Maldonado, and P. D. Tortell. 2016. Diurnal variation in the coupling of photosynthetic electron transport and carbon fixation in iron-limited phytoplankton in the NE subarctic Pacific. Biogeosciences 13: 1019–1035. doi:10.5194/bg-13-1019-2016

Zhao, Y., and A. Quigg. 2015. Study of photosynthetic productivity in the Northern Gulf of Mexico: Importance of diel cycles and light penetration. Continental Shelf Research 102: 33–46. doi:10.1016/j.csr.2015.04.014

Abstract:

"Also, the FRRf is still relatively novel,"

developed ~25 years ago, with the classic foundational papers appearing 1998...not really novel.

>>> We clarified the statement as follows: 

“Also, FRRf studies are limited in freshwater lakes where phosphorus limitation and cyanobacterial blooms are common.” (L30-31)

"The range of Ф e,C in the phytoplankton community varied from 1.1 to 31.0 mol e − mol C −1 during the study period"

Rather:

"The apparent range of Ф e,C in the phytoplankton community varied from 1.1 to 31.0 mol e − mol C −1 during the study period"

The actual ratio of e- : C has to be >= 4

>>> We corrected the sentence, as suggested (L36). Thank you.

References:

There is a newer paper by Max Gorbunov appearing in MEPS that is highly relevant.

>>> Thank you for the updated information. We introduced Gorbunov and Falkowski 2020 in discussion (L666-669) as follows,

“Further, a recent study reported that net primary productivity can also be measured by FRRf when fluorescence induction and relaxation (FIRe) techniques are applied (Gorbunov and Falkowski). This fact suggests that the application of FRRf enables real-time measurements of the PSII photochemical characteristics and the carbon metabolism of phytoplankton simultaneously.”

P.6

"Minimum PSII Fluorescence yield (under background light)"

Rather:

"Minimum PSII Fluorescence yield (under acclimation to background light)"

FO' is measured in darkness or back calculated to estimate fluorescence in darkness with C = 1

>>> We corrected Table 1, as suggested. (Table 1) Thank you.

P7

"Maximum photochemical efficiency under dark conditions"

No.

For cyanobacteria Fv/Fm is not an accurate proxy for Maximum photochemical efficiency under dark conditions", because of state transitions and non-PSII contributions to measured fluorescence.

Given the topic of the manuscript, using higher plant assumptions/short hand is not wise.

>>> Thank you for pointing this out. We clarified the definition of Fv/Fm in Table 1 as “(Fm-Fo)/Fm”. (Table 1)

P.8

mg vs. g vs. umol

Why not express everything on a umol basis, instead of flipping units back and forth?

It is a trivial issue, but raises a barrier for estimates like phi eC

>>> Yes, we agree with your comment. However, most studies about primary productivity consider the production rate based on the weight of carbon, such as g C/m2/d. Thus, we also described the oxygen productivity based on the weight for accessibility and comparative studies. Unit conversions are trivial, as suggested, but we believe this step-by-step description is better for non-expert (non bio-optical) readers. We hope you can agree with this assessment.

p.8:

"However, FRRf tended to over- and under-estimate GPP compared to the 14C and 13C

73 methods, and the light-dark bottle and 18O methods, respectively [32]"

this sentence is unclear, and 'respectively' is almost always a bad idea.

Just list the pairwise comparisons.

>>> Thank you for the comment. We revised this sentence as follows,

“However, FRRf measurements tended to overestimate GPP compared with the 14C and the 13C methods, while underestimating them when compared with the light-dark bottle and the 18O methods (Regaudie-de-Gioux et al. 2014).” (L71-73)

P.9

"derived ETRPSII to the GPP rate"

Actually, the relevant comparison is JVPSII to GPP.

ETRPSII does not account for [RCII].

Same comment at line 88, a direct comparison of ETRPSII to GPP does not really make sense, because it does not account for [RCII], or more generally for biomass.

>>> Thank you for pointing this out. We clarified ETRPSII to JVf accordingly (L76, L87, L88). 

Table 2:

There is a widely known pattern of mid-day/early afternoon depression of photosynthesis, but the study compares morning FRRf measures to afternoon Carbon measures or light dark bottle measures.

>>> We addressed this comment above, and confirmed additional measurement in in-situ and in laboratory. Table 2 has been modified to account for this clarification.

Materials & Methods:

"When the

163 logarithmic slope of Kd significantly changed with depth, we calculated it for each layer (Table 2)"

I do not understand this sentence, given the definition of Kd and the preceding equation.

More explanation needed; was E0 reset to the level reaching the top of each depth layer?

>>> Thank you for the comment. If Kd changed significantly at 5 m, we calculated Kd from 0 to 5 m (E0=E at 0 m), and 5 m to deeper layer (E0= E at 5 m). We revised this sentence as:

“We determined the diffuse attenuation coefficient Kd (m−1) with an exponential function as follows:

 E_bot=E_top exp⁡(–K_d×dZ) (1)

where Ebot and Etop are incident PAR (μmol photon m−2 s−1) at the bottom and top of the sampling layer, respectively, and dZ is the thickness of the layer. When the logarithmic slope of Kd did not change with depth, it was calculated as EZ = E0 exp (–Kd×dZ), where E0 is PAR at 0 m. When the logarithmic slope of Kd significantly changed with depth, we divided the water column into two layers at the depth where Kd was changed at and calculated it for each layer (Table 2)” (L184-189)

"The dark chamber has black housing and piping with a pump to

171 ensure that samples are measured under complete dark after 1–2 s of dark adaptation"

1-2 s is enough time for many cyanos to go from State I (illuminated) to State II (dark).

This can affect the estimate:

qP (=(F ´−FO ´)/(Fm ´−FO ´)

because F' is measured at State I and FO' is measured at State II (lower).

Also, NPQNSV reflects different mechanisms in cyanos and in eukaryotes

>>> Thank you for the comment. According to previous studies, state transition in cyanobacteria takes second to minutes (Aspinwall et al. 2004; Joshua and Mullineaux 2004; Kirilovsky 2015), as suggested. But, we cannot distinguish the state of each cell. Thus, we added notification about the interpretation of qP and NPQNSV due to the taxon-specific biases in methods as follows:

“It should be noted that, according to previous studies, PSII fluorescence of cyanobacteria could be lower under dark conditions due to the lag time associated with the transition from an illuminated to a dark state (State 1 to State 2) expressed in seconds to minutes (Aspinwall et al. 2004; Joshua and Mullineaux 2004; Kirilovsky 2015). Thus, when cyanobacteria are dominant in a community, FO and FO′ could be underestimated (Campbell et al. 1998), and cause overestimation of qP but underestimation of NPQNSV and RCII.” (L243-247)

>>>We also added new discussion about the effect of cyanobacterial state transition on JVf estimation as follows:

 “Another explanation is underestimation of RCII concentration under illuminated conditions (equation 2). This can be caused by decrease of FO under dark conditions due to the changes in fraction of excitation energy distribution between PSI and PSII (state transition) (Campbell et al. 1998).” (L610-613) 

Aspinwall, C. L., M. Sarcina, and C. W. Mullineaux. 2004. Phycobilisome mobility in the Cyanobacterium Synechococcus sp. PCC7942 is influenced by the trimerisation of photosystem I. Photosynthesis Research 79: 179. doi:10.1023/B:PRES.0000015399.43503.95

Campbell, D., V. Hurry, A. K. Clarke, P. Gustafsson, and G. Öquist. 1998. Chlorophyll fluorescence analysis of cyanobacterial photosynthesis and acclimation. Microbiol. Mol. Biol. Rev. 62: 667–683. doi:10.1128/MMBR.62.3.667-683.1998

Joshua, S., and C. W. Mullineaux. 2004. Phycobilisome diffusion is required for light-state transitions in cyanobacteria. Plant Physiology 135: 2112–2119. doi:10.1104/pp.104.046110

Kirilovsky, D. 2015. Modulating energy arriving at photochemical reaction centers: orange carotenoid protein-related photoprotection and state transitions. Photosynth Res 126: 3–17. doi:10.1007/s11120-014-0031-7

Line 213 and elsewhere:

There are minor grammar errors scattered through the text, and I apologize I am too pressed to note them all.

>>> Our revised manuscript has been checked and edited by English editing service. We will triple check, accordingly. Thank you.

Line 215: as noted above, back to umol e-, why mess around with g & mg O2 etc., Just use umol.

Line 227, back to mg O2... unnecessary flips, confusing conversion factors...

>>> As we noted above, this is bit confusing, but helpful information for non FRRf (nonexperts) and traditional primary production studies.

Line 237 etc.

"saturation phase under dark (RσPSII) and ambient light (RσPSII′)"

If you reject every measurement where RsigmaPSII' at the zeroth flashlet is > 0.08, you are going to reject many (most?) of the measures taken under illumination, since even low illumination will close 5-10% of RCII.

>>> All of opening RCII under illumination should be closed during the 100 flashlets of FRR fluorometer. In this context, already closed RCII should not re-open during the flashlets. Too high excitation flux will close RCIIs too quickly (RσPSII>0.08) which can result incorrect σPSII estimation. We adjusted excitation flux by Fastpro8 software to obtain optimal light-fluorescence curve.

Lines 256-260 would be better replaced by a table

>>> We created new table (Table 3). (L309)

Line 323

Line 333 "As in equation (8), Фe,C is defined as Jf /PBc × 43.2. "

another unnecessary and confusing unit conversion

>>> As we noted above, this is bit confusing, but acceptable for non experts. 

"Explanatory variables were standardized (mean 0 and standard

341 deviation 1) after log-transformation."

Hmmm.

The dynamic range vs. SD of the different metrics would be quite different; is it reasonable to force them all to the same scale?

>>> Standardization of explanatory variables is commonly used to know which variable is the most effective to response variable in a model. Because we measured all explanatory variables in the same day, the variance of those must reflect annual variation equally and be comparable.

Lines 350-371

This passage is highly phenomenological, without reference to the well understand patterns of light capture by cyanobacteria vs. other taxa.

Cyanos have a very small chl bed serving PSII, so very small sigmaPSII445nm compared to eukaryotes with much more chl serving PSII.

This is all well documented, example, Simis et al. Photosynthesis Research

>>> Thank you for the comment. We added new discussion (and references) about group-specific light harvest pigments and light capture as follows:

“It is well known that the optimal excitation waveband of cyanobacteria is between 550 and 650 nm, while that of eukaryotic algae is between 400 and 550 nm (Schubert et al. 1989; Simis et al. 2012). Therefore, the blue excitation flash at 444 nm can fail to saturate the RCII in cyanobacteria during a single turnover measurement of FRRf (Raateoja et al. 2004; Suggett et al. 2009b), and thus underestimate Fo (Simis et al. 2012) and GPP (Suggett et al. 2001) due to mismatch in wavelength between excitation wavelengths of the FRRf and the absorption spectrum of cyanobacteria. (L527-532)

Raateoja, M., J. Seppälä, and P. Ylöstalo. 2004. Fast repetition rate fluorometry is not applicable to studies of filamentous cyanobacteria from the Baltic Sea. Limnology and Oceanography 49: 1006–1012. doi:10.4319/lo.2004.49.4.1006

Schubert, H., U. Schiewer, and E. Tschirner. 1989. Fluorescence characteristics of cyanobacteria (blue-green algae). J Plankton Res 11: 353–359. doi:10.1093/plankt/11.2.353

Simis, S. G. H., Y. Huot, M. Babin, J. Seppälä, and L. Metsamaa. 2012. Optimization of variable fluorescence measurements of phytoplankton communities with cyanobacteria. Photosynthesis Research 112: 13–30. doi:10.1007/s11120-012-9729-6

Suggett, D. J., C. M. Moore, A. E. Hickman, and R. J. Geider. 2009. Interpretation of fast repetition rate (FRR) fluorescence: signatures of phytoplankton community structure versus physiological state. Marine Ecology Progress Series 376: 1–19. doi:10.3354/meps07830

Lines 413-415:

" we used the

414 data set that was obtained by the combination of three excitation wavelengths due to the quality and

415 reliability (Fig 2, 3). "

But, given the changes in cyano dominance, would it better to apply a spectral regime weighted to the community composition?

>>> As shown in Table 4, the data quantity and quality was the best when we used three excitation wavelengths combination. Thus, we used these data set for analyses regardless the phytoplankton composition in sampling site.

Line 428 etc.

Before applying an explanatory GLM, would it be wise to screen (or correct) the estimates of phi eC that are lower than theoretically possible?

Explaining measures that must be wrong seems odd.

>>> Although Фe,C <4 are theoretically wrong, we believe that the variance of Фe,C is an important information, as in previous study (Lawrenz et al. 2013). Actually, rejection of smaller values did not affect our conclusion: We tested new data set which excluded Фe,C <4 on GLM analysis and found that although coefficient of Chl-a became second highest, temperature was still the most important factor while PAR and PO4 were less important. 

Variables Coefficient Std. Error t value P

Best model, AIC=11754, R2=0.71 　 　 　

Intercept 2.08 0.01 360.2 < 2e-16

PAR -0.04 0.01 -4.5 7.96E-06

Temperature 0.49 0.02 30.1 < 2e-16

DO -0.17 0.01 -14.0 < 2e-16

Turbidity -0.03 0.01 -2.4 0.015342

Fv/Fm -0.13 0.01 -14.8 < 2e-16

σPSII 0.05 0.01 6.2 5.67E-10

NH4 -0.09 0.01 -11.2 < 2e-16

NO2+NO3 0.18 0.01 22.2 < 2e-16

PO4 -0.02 0.01 -3.6 0.000288

Chl-a 0.41 0.02 27.0 < 2e-16

Cyanobacteria -0.29 0.01 -28.6 < 2e-16

Diatoms -0.27 0.01 -25.8 < 2e-16

Cryptophytes -0.27 0.01 -25.765 < 2e-16

Lawrenz, E., G. Silsbe, E. Capuzzo, and others. 2013. Predicting the electron requirement for carbon fixation in seas and oceans L.J. Stal [ed.]. PLoS ONE 8: e58137. doi:10.1371/journal.pone.0058137

Line 438:

"The effects of PAR and diatoms in

438 this analysis may have included those of NPQNSV and zygnematphytes."

This sentence does not make sense as written. 

>>>We rewritten this sentence as: 

“The explanatory variable ‘PAR’ may include influences of both PAR and NPQNSV in the GLM due to the high correlation between the two. Similarly, the explanatory variable ‘diatoms’ in this analysis may include influences of both diatoms and zygnematophytes.” (L494-497).

Line 482:

"In this study, Фe,C in Lake Biwa ranged temporally from 1.1 to 31.0 mol e− mol C−1 (Fig. 4)."

Rather:

In this study, apparent Фe,C in Lake Biwa ranged temporally from 1.1 to 31.0 mol e− mol C−1 (Fig. 4)."

Фe,C cannot actually go below 4, so measures <4 are telling us about issues with approach. 

>>> We fixed that as it pointed out. Also, we added new discussion as follows.

“The smaller than theoretical Фe,C values can be due to underestimation of JVf especially when cyanobacteria dominated (Simis et al. 2012; Robinson et al. 2014) and/or overestimation of carbon fixation rate due to bottle effects (Suggett et al. 2009a; Lawrenz et al. 2013; Zhu et al. 2017). The former case is plausible because the GLM revealed that the effect of cyanobacteria on Фe,C was significantly negative (see below). The latter case can be more likely in this study, due to mitigation of UV stresses in the growth chamber and an increasing growth rate of cells on bottle walls (Venrick et al. 1977; Carpenter and Lively 1980). “ (L561-566)

Carpenter, E. J., and J. S. Lively. 1980. Review of Estimates of Algal Growth Using 14C Tracer Techniques, p. 161–178. In P.G. Falkowski [ed.], Primary Productivity in the Sea. Springer US.

Venrick, E. L., J. R. Beers, and J. F. Heinbokel. 1977. Possible consequences of containing microplankton for physiological rate measurements. Journal of Experimental Marine Biology and Ecology 26: 55–76. doi:10.1016/0022-0981(77)90080-6

Line 491:

"The NPQNSV is mechanistically

492 linked with alternative electron flow (AEF) activity, which is activated by excess light and

493 photodamage on PSII [1,64,97]. "

Rather:

"The NPQNSV is phenomenologically correlated

492 linked with alternative electron flow (AEF) activity, which is activated by excess light and

493 photodamage on PSII [1,64,97]. "

Both NPQ and AEF are induced under conditions of excess light; AEF can then loop back to induce NPQ.

>>> We clarified this sentence as suggested (L570). Thank you. 

Reviewer #2: This is a technically well executed, and well-interpreted, manuscript.

A few points need attention.

Line 54. ‘rapidly adapt’: does this mean adapt in the strict (and preferable) sense of genetic change plus natural selection, or in the more general sense of genetic adaptation plus phenotypic acclimation. The citation of Reynolds (1976) post-dates the initial paper of Collins and Bell (1974; Nature 431, 566-569) on experimental evolution of phytoplankton giving an estimate of the time over which genetic adaptation can occur.

>>> Thank you for the comment. We changed ‘adapt’ to ‘respond’ to avoid confusion. (L50)

Lines 79-81. Is the increased light availability in the open ocean than coastal waters a result of increased phytoplankton per unit volume in coastal waters, higher concentrations of gelbstoff, more suspended non-living particles, or all three? If it is primarily phytoplankton density, is the total light available in the euphotic zone (less deep in the coastal ocean) less than in coastal waters?

>>>>The low light availability in coastal waters can be caused by all three factors but Lawrenz et al. (2013) did not discussed which is the main factor affecting light environment among study area. Also, vertical distribution of phytoplankton is markedly different in open-ocean and coastal areas, where the former is located much deeper (and less gelbstoff and non-algal) in the water column - thus, technically more light availability. On the other hand, light availability (for photosynthesis) could also be theoretically high in coastal areas through increased scattering in the euphotic layer when more particles and non-algal matter are high in concentration. This is an interesting hypothesis that should be addressed in future studies.

Lawrenz, E., G. Silsbe, E. Capuzzo, and others. 2013. Predicting the electron requirement for carbon fixation in seas and oceans L.J. Stal [ed.]. PLoS ONE 8: e58137. doi:10.1371/journal.pone.0058137

Line 82. Admittedly ‘such as’ is used for the Mehler reaction, into would be useful to mention flavodi-iron proteins and PTOX.

>>> We modified the sentence, as suggested. Thank you.

“More specifically, excess light energy enhances photo-oxidative damage and alternative electron transport such as the Mehler reaction, flavodi-iron protein-mediated electron flows, and plastiquinol terminal oxidase (PTOX) pathway, which can increase Фe,C (Roach and Krieger-Liszkay 2014; Hughes et al. 2018).” (L82-83)

Lines 92-94. Good point.

Lines 104-105. Is ATP availability the main effect of P limitation. What about decreased levels of the three main forms of RNA (rRNA, mRNA, tRNA) that restrict the rate of protein synthesis?

>>> ATP availability affects directly photosynthesis. RNA is also a P-rich molecule, and cellular RNA content correlated with growth rates. Previous study indicated that P-limitation for photosynthetic organisms do not always reduce the cellular RNA level (Raven 2013). 

Raven, J. A. 2013. RNA function and phosphorus use by photosynthetic organisms. Front. Plant Sci. 4. doi:10.3389/fpls.2013.00536

Line 110, 117. Is this P in the water column, or in cells.

>>> We added “in the water column” accordingly. (L113, 120)

Line 151. No method is given for measuring ammonium, but ammonium concentrations are mentioned on (e.g.) line 392.

>>> NH4 was also analyzed by ion chromatograph system. We added “ammonia” in method. (L170)

Line 246. ‘chrysophytes’, not ‘crysophytes’. Also, do the chrysophytes include the Synurophyceae as well as the Chrysophyceae”

>>> We fixed as ‘chrysophytes’. (L288). But Synurophyceae is separated from Chrysophyceae (https://www.algaebase.org/browse/taxonomy/?id=4339).

Line 352. ‘to a cyanobacterial’

>>. We fixed that as it pointed out. Thank you (L403)

Lines 498-500. Clarify the effects of temperature and oxygen on Rubisco kinetic properties (Galmes et al. 2014 Photosynthesis Research 123: 183-201). Also, while low fluxes through glycolate metabolism is essential for cyanobacteria (Eisenhut et al. 2008 PNAS 106: 17199-17204), carbon concentrating mechanisms restrict Rubisco oxygenase activity and hence photorespiration in air-equilibrium solutions for cyanobacteria and many eukaryotic microalgae (Raven et al. 2017 Journal of Experimental Botany 68: 3701-3716), and even at the increased carbon dioxide concentrations in some freshwaters as a result of carbon dioxide from soil respiration in water entering the lake, as well mineralisation of organic matter entering lakes from the catchment.

>>> Thank you for the comment. We modified the discussion, as suggested.

“Although the temperature effects on Фe,C were not specifically addressed, the results from the present study can discuss mechanisms related to the relationship. First, increased temperature decreases the CO2 affinity of Rubisco through the acceleration of the O2 evolution rate, and reduction of CO2 solubility (Ogren 1984; Hagemann et al. 2016). CO2 selectivity of Rubisco depends on its form (Iñiguez et al. 2020): Form IA Rubisco (cyanobacteria) and form IB Rubisco (cyanobacteria and chlorophytes) have lower selectivity for CO2 than form ID Rubisco (diatoms). In Lake Biwa, cyanobacteria and zygnematophytes were dominant groups during summer and autumn (S6 Appendix). Thus, Rubisco CO2 affinity of phytoplankton community in Lake Biwa might be decreased following the increasing of water temperature and enhanced Rubisco oxygenation (photorespiration). Second, to mitigate photorespiration, phytoplankton express CO2 concentration mechanisms (CCMs) (Giordano et al. 2005). CCMs need ATP or NADPH that are also needed for driving the Calvin-Benson cycle. Finally, increasing temperature may lead to a state of chronic photoinhibition through photodamage (Yellowlees and Warner 2003), as well as changes in species composition of the community (Ryan-Keogh et al. 2018). Thus, although we did not examine the interaction among all explanatory variables, temperature may have affected Фe,C vis-à-vis interaction with the other factors such as nutrient stoichiometry (Fernández‐González et al. 2020) “(L578-592).

Giordano, M., J. Beardall, and J. A. Raven. 2005. CO 2 concentrating mechanisms in algae: mechanisms, environmental modulation, and evolution. Annu. Rev. Plant Biol. 56: 99–131. doi:10.1146/annurev.arplant.56.032604.144052

Iñiguez, C., S. Capó‐Bauçà, Ü. Niinemets, H. Stoll, P. Aguiló‐Nicolau, and J. Galmés. 2020. Evolutionary trends in RuBisCO kinetics and their co-evolution with CO2 concentrating mechanisms. The Plant Journal 101: 897–918. doi:10.1111/tpj.14643

Reviewer #3: This is a very thorough investigation of alternative methods for measuring lake production. A high level of detail is provided, and it should be possible for other groups to reference this work and use the same comparison methods in other lakes. The results add to the growing body of evidence which is showing that the relationship between the quantum yields of electron transport and that for carbon fixation is highly variable, but that a good proportion of the variability can be explained by the growth conditions of the phytoplankton. Even more encouraging from a remote sensing point of view, is that simple to measure variables such as lake temperature are weighted highly in the model outcomes.

I have one caveat before publication, however, and it needs consideration. A large set of biooptical measurements are presented here, but a critical variable is missing, and that is the phytoplankton absorption spectrum. Here, the spectrum is modelled rather than measured using Paavel 2016. This could bring a large uncertainty into the results, if the cyanobacterial-domitaed community of Lake Biwa is different from the lakes used by Paaval. This is one point to be discussed - are the modelled absorption spectra realistic?

>>> Thank you for the valuable comment. The short answer is, Yes. Absorption spectra (a*phy) of mixed algal community might differ from pure culture due to the pigment composition and package effect. In Lake Biwa, Anabaena affinis and Aphanothece sp. dominated >90% of the cyanobacterial biomass at 0 and 2 m in August. The former species has phycocyanin (PC) (Wojtasiewicz and Stoń-Egiert 2016) while later has PC and phycoerythrin (PE) (Hirose et al. 1969; Rohatgi and Singh 1979). In Paavel’s study, summer cyanobacterial bloom consisted of Anabaena, Aphanizomenon and Gloeotrichia (Paavel et al. 2016). The former two species have PC and later species has PC and PE. We tested a*phy of pure Anabaena culture (Wojtasiewicz and Stoń-Egiert 2016) for August in Lake Biwa, but SCF changed only 3%. Even though, we think using Paavel’s a*phy coefficients might be better for the reason mentioned above.

Hirose, H., S. Kumano, and K. Madono. 1969. Spectroscopic studies on phycoerythrins from cyanophycean and rhodophycean algae with special reference to their phylogenetical relations. Bot Mag Tokyo.

Paavel, B., K. Kangro, H. Arst, A. Reinart, T. Kutser, and T. Nõges. 2016. Parameterization of chlorophyll-specific phytoplankton absorption coefficients for productive lake waters. Journal of Limnology 75. doi:10.4081/jlimnol.2016.1426

Rohatgi, A., and S. P. Singh. 1979. Isolation and characterization of pigment mutants of the blue-green alga Aphanothece stagnina. Molec. Gen. Genet. 169: 59–62. doi:10.1007/BF00267545

Wojtasiewicz, B., and J. Stoń-Egiert. 2016. Bio-optical characterization of selected cyanobacteria strains present in marine and freshwater ecosystems. J Appl Phycol 28: 2299–2314. doi:10.1007/s10811-015-0774-3

A second point is a broader critique on the use of absorption for modelling the photosynthetic response. In groups such as cyanobacteria and red algae, there is very little photosynthetic oxygen evolution driven by the blue end of the spectrum. This is due to the completely different antenna of PSII and PSI in these groups. Most of the chlorophyll, and hence blue absorption is associated with PSI. For this reason, it is more accurate to use the action spectrum instead of the absorption spectrum, and this point is made many times in the literature (e.g. Johnsen et al 2007, reference 92). Not many labs are able to measure the PS2 action spectrum, and this (correct) method is not so widely used compared to the default of using absorption. The difference is not so great for diatom-dominated waters, as absorption and action cpectra are rather similar, but there are large differences for cyanos - both marine and freshwater. As the owner of a blue-light only FastTracka myself, I am very aware of this problem. I would like to see a couple of sentences in the discussion mentioning that action spectra could be a better alternative.

>>> Thank you for the important suggestion. We added discussion as suggested.

“Johnsen et al. (Johnsen and Sakshaug 2007) suggested that the fraction of light absorption by PSII, or that of cellular Chl-a in PSII, can be a useful proxy as action spectra, or alternative to absorption spectra, to correct the effect of photoprotective carotenoids and PSI-PSII absorption variability. Considering the large differences in regulation of the distribution of light harvesting pigments and excitation energy between cyanobacteria and eukaryotic algae, the action spectra proxies should provide more accurate measurement of GPP in future study. Further, evaluating the proportion of cyanobacteria in phytoplankton biomass will be crucial for correcting GPP estimation with ETRPSII in natural phytoplankton communities.” (L614-620)

Johnsen, G., and E. Sakshaug. 2007. Biooptical characteristics of PSII and PSI in 33 species (13 pigment groups) of marine phytoplankton, and the relevance for pulse-amplitude-modulated and fast-repetition-rate fluorometry. Journal of Phycology 43: 1236–1251. doi:10.1111/j.1529-8817.2007.00422.x

I have not made a detailed file of minor corrections on the ms. due to lack of time, but the pdf attached has a few changes.

>>> Thank you for the comments. We fixed those as follows,

“A generalized linear model...best fit” (L37-38)

 “Moreover, the blue light (< 500 nm) absorbed by cyanobacterial antenna pigments does not make a major contribution to the reduction of the QA and PQ pool, and O2 evolution (Subramaniam et al. 1999)”. (L98-100)

Subramaniam, A., E. J. Carpenter, D. Karentz, and P. G. Falkowski. 1999. Bio-optical properties of the marine diazotrophic cyanobacteria Trichodesmium spp. I. Absorption and photosynthetic action spectra. Limnology and Oceanography 44: 608–617. doi:10.4319/lo.1999.44.3.0608

---

## [Decision Letter · Decision Letter 1]

8 Dec 2020

PONE-D-20-24363R1

Development of photosynthetic carbon fixation model using multi-excitation wavelength fast repetition rate fluorometry in Lake Biwa

PLOS ONE

Dear Dr. Kazama,

Thank you for submitting your manuscript to PLOS ONE. After careful consideration, we feel that it has merit but does not fully meet PLOS ONE’s publication criteria as it currently stands. Therefore, we invite you to submit a revised version of the manuscript that addresses the points raised during the review process.

Overall, you did a very good job addressing the reviewers comments. However reviewer 1 still makes a very good point that the analysis returns impossible values and that the manuscript should be very clear about that issue. Please read careful the reviewer's comments and include them in the final version.

Personally, I would like to see some of the other reviewers original comments addressed in the manuscript. Namely:

Reviewer 2

The comment on L79-81 of the original manuscript was answered on your review but nothing was added to the manuscript, please do so. The same for L104-105 comment.

Reviewer 3

Your answer to the first comment merits being added to the M&M.

We look forward to receiving your revised manuscript.

Kind regards,

Bruno Jesus, Ph.D

Academic Editor

PLOS ONE

Reviewers' comments:

Reviewer's Responses to Questions

**Comments to the Author**

1. If the authors have adequately addressed your comments raised in a previous round of review and you feel that this manuscript is now acceptable for publication, you may indicate that here to bypass the “Comments to the Author” section, enter your conflict of interest statement in the “Confidential to Editor” section, and submit your "Accept" recommendation.

Reviewer #1: (No Response)

2. Is the manuscript technically sound, and do the data support the conclusions?

Reviewer #1: Partly

3. Has the statistical analysis been performed appropriately and rigorously? 

Reviewer #1: I Don't Know

4. Have the authors made all data underlying the findings in their manuscript fully available?

Reviewer #1: Yes

5. Is the manuscript presented in an intelligible fashion and written in standard English?

Reviewer #1: Yes

6. Review Comments to the Author

Reviewer #1: The author's responses have largely addressed my comments and concerns.

I remain concerned about predictive modelling based upon impossible data. Unless something deeply unexpected is happening phi e, C cannot be below 4.

Yet many/most of the estimated values for phi e, C are below 4.

So ,I think it should be crystal clear, throughout, that the authors have applied relatively standard approaches, and, consistent with previous attempts, are getting mechanistically impossible answers. So, the field is, consistently, generating paradoxical findings which most likely reflect core issues with the approaches.

And, I am still concerned about the temporal offset between the in situ FRRf numbers and the in lab CO2 uptake numbers, which would go some way towards addressing the inconsistency, I think.

The morning in situ FRRf numbers are likely before any induction of community level photoinhibition.

The afternoon in lab CO2 uptake numbers may be after the onset of community level photoinhibition, or possibly even mid-day limitation on available DIC.

But, as the authors point out, if morning FRRf ETR is exaggerated relative to afternoon CO2 uptake, this would push phi e, C up, not down.

Specific Comments

Abstract:

"The GPP values estimated by FRRf ( GPP f ) with the best Ф e,C model relative to 13 C ( GPP 13C ) varied in the range of 0.5–1.5."

This is possibly predictively useful, but mechanistically impossible.

It is worth mentioning that Ф e,C cannot, mechanistically, be less than 4, and is more likely 5 or above.

"Thus, when cyanobacteria are dominant in a community, FO and FO′ could be underestimated (Campbell et al. 1998), "

Actually, when cyanobacteria are dominant it leads to an over-estimation of PSII derived FO, because phycobiliproteins contribute to measured F0.s

7. PLOS authors have the option to publish the peer review history of their article (what does this mean?). If published, this will include your full peer review and any attached files.

Reviewer #1: **Yes: **Douglas A. Campbell

---

## [Author Response · Author response to Decision Letter 1]

6 Jan 2021

Editor comments

Personally, I would like to see some of the other reviewers original comments addressed in the manuscript. Namely:

Reviewer 2

The comment on L79-81 of the original manuscript was answered on your review but nothing was added to the manuscript, please do so. The same for L104-105 comment.

>>> We revised the manuscript to match our reply as follows,

“Light availability can be affected by the concentrations of colored dissolved organic matter (CDOM), non-algal particles, and phytoplankton. Moreover, the vertical distribution of phytoplankton is markedly different in open-ocean and coastal areas, where the former has peak biomass located in a much deeper area with less CDOM and non-algal particles in the water column, allowing more light availability [35].” (L79-83)

>>> L104-105 was revised as follows,

“Whereas nitrogen limitation depresses cellular Chl-a concentration, phosphorus limitation inhibits RNA and ATP synthesis, which can affect protein synthesis and photochemical energy conversion in algae [56, 57].” (L110-111)

Reviewer 3

Your answer to the first comment merits being added to the M&M.

>>> We included the following to the M&M in L343-349.

“In Lake Biwa, Anabaena affinis and Aphanothece sp. dominated >90% of the cyanobacterial biomass at 0 and 2 m, respectively, in August 2018. The former species has phycocyanin (PC) [49], whereas the latter has PC and phycoerythrin (PE) [88,89]. Paavel et al. [86] described a summer cyanobacterial bloom that consisted of Anabaena, Aphanizomenon, and Gloeotrichia. The former two species have PC [49], and later species has PC and PE [90]. We tested the a_phy^* (λ) of pure Anabaena [49] culture in August 2018 in Lake Biwa, but the SCF changed to only 3%. We believe that Paavel et al. a_phy^* (λ) coefficients are better for the abovementioned reason.”

>>>In addition, we checked all manuscript through English editing service, and fixed following words.

analyser → analyzer (L176&320)

categorised → categorized (L291)

Reviewer #1: The author's responses have largely addressed my comments and concerns.

I remain concerned about predictive modelling based upon impossible data. Unless something deeply unexpected is happening phi e, C cannot be below 4.

Yet many/most of the estimated values for phi e, C are below 4.

So, I think it should be crystal clear, throughout, that the authors have applied relatively standard approaches, and, consistent with previous attempts, are getting mechanistically impossible answers. So, the field is, consistently, generating paradoxical findings which most likely reflect core issues with the approaches.

>>> Thank you for pointing this out. We added the some notices about unrealistic Фe,C values (<4) in abstract and discussion as follows,

“This study quantifies the in situ FRRf method in a freshwater ecosystem, discusses core issues in the methodology to calculate Фe,C, and assesses the applicability of the method for lake GPP prediction.”(L41-42)

“It is also possible that the geometric properties of the incubation vessel can alter the light environment on each bottle, and cause calculation errors in ETRPSII and underestimation of Фe,C [43]. In any case, although we applied relatively standard methods throughout the study, i.e., FRRf vs. bottle incubations with an isotopic tracer, the results showed Фe,C < 4, which may reflect core issues in this approach.

The Фe,C values < 4 have often been observed in laboratory cultures [43,70], natural communities in open oceans [26,40,44], and coastal regions [25,36,39]. However, the analysis method for the low Фe,C values is not yet unified. Most of the studies included the low Фe,C values in the statistical analyses [36,39,40,44,70], while a few studies did not [25,26]. In this study, we regarded all of the Фe,C values as apparent values as in the previous study [35] since the bottle incubation method can cause artifacts for all bottles. Thus, the low Фe,C values were included in the GLM analysis to clarify the apparent effect of environmental and biological factors on the variance of Фe,C.” (L575-586)

“These results must be interpreted with caution because although Фe,C should be mechanistically ≥ 4 mol e− mol C−1, our study showed Фe,C < 4 in 9 out of 14 observations. In this study, we define the observed Фe,C as apparent values that are included artifacts from methodological issues, such as bottle effects, which can be minimized by a relatively shorter incubation time (~2 h) [122]. Moreover, studies are needed to develop the incubation-free techniques, such as cellular NADP(H) measurement [123].” (L680-685) 

122. Milligan AJ, Halsey KH, Behrenfeld MJ. Advancing interpretations of 14C-uptake measurements in the context of phytoplankton physiology and ecology. J Plankton Res. 2015;37: 692–698. doi:10.1093/plankt/fbv051

123. Jones BM, Halsey KH, Behrenfeld MJ. Novel incubation-free approaches to determine phytoplankton net primary productivity, growth, and biomass based on flow cytometry and quantification of ATP and NAD(H). Limnology and Oceanography: Methods. 2017;15: 928–938. doi:10.1002/lom3.10213

And, I am still concerned about the temporal offset between the in situ FRRf numbers and the in lab CO2 uptake numbers, which would go some way towards addressing the inconsistency, I think.

The morning in situ FRRf numbers are likely before any induction of community level photoinhibition.

The afternoon in lab CO2 uptake numbers may be after the onset of community level photoinhibition, or possibly even mid-day limitation on available DIC.

But, as the authors point out, if morning FRRf ETR is exaggerated relative to afternoon CO2 uptake, this would push phi e, C up, not down.

>>> We agree with your comment. We modified the sentence to clarify the mechanisms of the diel variation of phytoplankton productivity as follows,

“Previous studies reported that the carbon fixation rate of natural communities is ~1.3 times higher in the morning than in the afternoon because photoinhibition and nutrients depletion suppress community productivity in the afternoon [34,109]. If we underestimated carbon fixation rates of the communities incubated in the growth chamber, the results might have overestimated Фe,C. However, Фe,C values were paradoxically smaller than theoretical values (4 mol e− mol C−1) during most of the observations (Fig.4), except on 28 August, 13 September … ”(L563-567) 

Specific Comments

Abstract:

"The GPP values estimated by FRRf ( GPP f ) with the best Ф e,C model relative to 13 C ( GPP 13C ) varied in the range of 0.5–1.5."

This is possibly predictively useful, but mechanistically impossible.

It is worth mentioning that Ф e,C cannot, mechanistically, be less than 4, and is more likely 5 or above.

>>> We inserted the sentence in abstract as follows,

“~carbon fixation (Фe,C), which is mechanistically 4 mol e− mol C−1 or above, can vary depending on multiple factors.” (L30)

"Thus, when cyanobacteria are dominant in a community, FO and FO′ could be underestimated (Campbell et al. 1998), "

Actually, when cyanobacteria are dominant it leads to an over-estimation of PSII derived FO, because phycobiliproteins contribute to measured F0.s

>>> Thank you for pointing this out. We corrected the sentences as suggested (L251-252).

---

## [Editor Report · Decision Letter 2]

20 Jan 2021

Development of photosynthetic carbon fixation model using multi-excitation wavelength fast repetition rate fluorometry in Lake Biwa

PONE-D-20-24363R2

Dear Dr. Kazama,

We’re pleased to inform you that your manuscript has been judged scientifically suitable for publication and will be formally accepted for publication once it meets all outstanding technical requirements.

Kind regards,

Bruno Jesus, Ph.D

Academic Editor

PLOS ONE
---

## [Editor Report · Acceptance letter]

22 Jan 2021

PONE-D-20-24363R2 

Development of photosynthetic carbon fixation model using multi-excitation wavelength fast repetition rate fluorometry in Lake Biwa 

Dear Dr. Kazama:

I'm pleased to inform you that your manuscript has been deemed suitable for publication in PLOS ONE. Congratulations! Your manuscript is now with our production department. 

Kind regards, 

on behalf of

Dr. Bruno Jesus 

Academic Editor

PLOS ONE